# Xanomeline displays concomitant orthosteric and allosteric binding modes at the M$_4$ mAChR

Wessel A. C. Burger [1,2,9], Vi Pham[1,9], Ziva Vuckovic[1,9], Alexander S. Powers[3,4,9], Jesse I. Mobbs [1,2], Yianni Laloudakis[3], Alisa Glukhova [1,2], Denise Wootten [1,2], Andrew B. Tobin [5], Patrick M. Sexton [1,2], Steven M. Paul[6], Christian C. Felder[6], Radostin Danev [7], Ron O. Dror [4] ✉, Arthur Christopoulos [1,2,8] ✉, Celine Valant [1] ✉ & David M. Thal [1,2] ✉

The M$_4$ muscarinic acetylcholine receptor (M$_4$ mAChR) has emerged as a drug target of high therapeutic interest due to its expression in regions of the brain involved in the regulation of psychosis, cognition, and addiction. The mAChR agonist, xanomeline, has provided significant improvement in the Positive and Negative Symptom Scale (PANSS) scores in a Phase II clinical trial for the treatment of patients suffering from schizophrenia. Here we report the active state cryo-EM structure of xanomeline bound to the human M$_4$ mAChR in complex with the heterotrimeric G$_{i1}$ transducer protein. Unexpectedly, two molecules of xanomeline were found to concomitantly bind to the monomeric M$_4$ mAChR, with one molecule bound in the orthosteric (acetylcholine-binding) site and a second molecule in an extracellular vestibular allosteric site. Molecular dynamic simulations supports the structural findings, and pharmacological validation confirmed that xanomeline acts as a dual orthosteric and allosteric ligand at the human M$_4$ mAChR. These findings provide a basis for further understanding xanomeline's complex pharmacology and highlight the myriad of ways through which clinically relevant ligands can bind to and regulate GPCRs.

Schizophrenia is a debilitating and complex psychiatric disease that affects ~1% of the global population[1]. Current frontline treatments for schizophrenia are generally classified as 'typical' and 'atypical' antipsychotics that primarily antagonise dopamine D$_2$ receptors along with other G protein-coupled receptors (GPCRs) that exhibit complex polypharmacology and undesirable metabolic, cognitive, and motor side effects that limit therapy[2]. Consequently, almost 70% of schizophrenia patients discontinue their treatment within the first 18 months[3].

The five human muscarinic acetylcholine receptors (M$_1$–M$_5$ mAChRs) are Class A GPCRs that are widely expressed throughout the

[1]Drug Discovery Biology, Monash Institute of Pharmaceutical Sciences, Monash University, Parkville, VIC 3052, Australia. [2]Australian Research Council Centre for Cryo-Electron Microscopy of Membrane Proteins, Monash Institute of Pharmaceutical Sciences, Monash University, Parkville, VIC 3052, Australia. [3]Department of Chemistry, Stanford University, Stanford, CA 94305, USA. [4]Departments of Computer Science, Structural Biology, and Molecular and Cellular Physiology, Stanford University, Stanford, CA 94305, USA. [5]The Advanced Research Centre (ARC), Centre for Translational Science, School of Biomolecular Sciences, College of Medical, Veterinary and Life Sciences, University of Glasgow, Glasgow G12 8QQ, UK. [6]Karuna Therapeutics, Boston, MA 02110, USA. [7]Graduate School of Medicine, University of Tokyo, N415, 7-3-1 Hongo, Bunkyo-ku 113-0033 Tokyo, Japan. [8]Neuromedicines Discovery Centre, Monash University, Parkville, VIC 3052, Australia. [9]These authors contributed equally: Wessel A. C. Burger, Vi Pham, Ziva Vuckovic, Alexander S. Powers. ✉e-mail: ron.dror@stanford.edu; arthur.christopoulos@monash.edu; celine.valant@monash.edu; david.thal@monash.edu

central nervous system (CNS) and peripheral nervous system and have been implicated in the regulation of cognition, psychosis, and addiction[4]. As a result, the mAChRs have become therapeutic targets for the treatment of numerous central nervous system disorders including Alzheimer's disease, schizophrenia, and drug addiction; currently, there are no FDA-approved mAChR ligands for the treatment of these disorders[4,5].

Xanomeline is an orthosteric mAChR agonist with clinical efficacy for the treatment of schizophrenia and Alzheimer's disease (AD)[6–8]. Unfortunately, xanomeline displayed dose-limiting side effects that prevented its clinical translation in AD. However, the therapeutic potential of xanomeline renewed interest in the development of safer mAChR therapeutics for this indication via improved mAChR subtype selectivity[9]. Consequently, multiple mAChR ligands are now progressing through Phase II and III clinical trials, including selective mAChR agonists[10], selective positive allosteric modulators (PAMs)[11], and in the case of xanomeline, a dual therapy treatment combining xanomeline with the peripherally restricted pan-mAChR antagonist trospium chloride (KarXT), which allows penetration of xanomeline into the CNS while reducing incidences of peripheral mAChR-mediated adverse events[12–14].

Historically, xanomeline has been described as a functionally "$M_1$ and $M_4$ mAChR preferring" agonist, albeit with equivalent affinity for the remaining mAChR subtypes and some affinity for the 5-$HT_1$ and 5-$HT_2$ serotonin receptors[15]. However, recent studies suggest that xanomeline's antipsychotic efficacy is due to preferential agonism at the $M_4$ mAChR[11]. For example, the antipsychotic efficacy of xanomeline was completely abolished in $M_4$ mAChR knockout (KO) mice, while only being partially reduced in $M_1$ mAChR KO mice[16]. In addition, xanomeline-mediated signalling at the $M_4$ mAChR in rodent brain occurred at significantly lower concentrations than at the $M_1$ mAChR[17].

As a consequence, understanding how xanomeline interacts with the therapeutically relevant $M_4$ mAChR subtype is key for designing future novel and more targeted first-in-class antipsychotics. In this study, we determined the single-particle cryo-EM structure of the active-state $M_4$ mAChR bound to xanomeline in complex with its cognate heterotrimeric G protein $G_{i1}$. Unexpectedly, the cryo-EM structure revealed two molecules of xanomeline concomitantly bound to the $M_4$ mAChR, with one molecule bound in the primary, orthosteric site, and a second xanomeline molecule bound in an allosteric site within the receptor's extracellular vestibule (ECV), which has previously been shown to be present on all mAChRs[18]. To verify the unappreciated potential of xanomeline binding to the mAChR allosteric site, we performed molecular dynamics (MD) simulations and molecular pharmacology experiments that validated this novel mode of target engagement. These findings provide new insights into the molecular pharmacology of xanomeline and can enable the structure-based design of novel selective mAChR ligands.

## Results
### Structure of xanomeline bound $M_4$ mAChR-$G_{i1}$ complex
To obtain a structure of the $M_4$ mAChR bound to xanomeline and transducer $G_{i1}$, we used methodology similar to that used for the determination of agonist (iperoxo) bound $M_1$, $M_2$, and $M_4$ mAChR complex structures (Supplementary Fig. 1)[19,20]. Purified xanomeline-bound $M_4$ mAChR-$G_{i1}$ (xano-$M_4$R-$G_{i1}$) complex was imaged by single-particle cryo-transmission electron microscopy (EM) on a Titan Krios microscope (Supplementary Table 1). We obtained a cryo-EM structure of the xano-$M_4$R-$G_{i1}$ complex at a global resolution of 2.5 Å (Supplementary Table 1, Supplementary Figs. 2–3). Cryo-EM maps allowed for placement of all the components of the complex including receptor, $G\alpha_{i1}\beta_1\gamma_2$, scFv16[21], and two molecules of xanomeline (Fig. 1a, Supplementary Fig. 3).

Overall, the xano-$M_4$R-$G_{i1}$ structure was similar to previous structures of iperoxo-bound (ipx) to the $M_1$ and $M_2$ mAChRs[19] with

root mean square deviations (RMSD) of 0.64 Å and 0.65 Å for the whole complex. The xano-$M_4$R-$G_{i1}$ structure was also similar to an ipx-bound structure of the $M_4$ mAChR that was determined by our group with a RMSD of 0.30 Å[20]. We note that this contrasts to another LY2119620-ipx-$M_4$R-$G_{i1}$ complex structure that has RMSD differences of 1.1 Å for the complexes and 0.85 Å for the receptors (Supplementary Fig. 4)[22]. The larger differences between the $M_4$ mAChR structures are likely not due to bona fide structural differences, but rather due to lower local resolution in the cryo-EM density maps and poor modelling of the structures to the maps (Supplementary Fig. 4)[23]. As such, comparisons of xano-$M_4$R-$G_{i1}$ were limited to our prior $M_4$ mAChR structures[20]. The G protein interface in our xano-$M_4$R-$G_{i1}$ was similar to that observed in previous agonist-bound $M_4$R-$G_{i1}$ structures, as well as other mAChR $G_{i/o}$ complex structures[19,20,22].

### Xanomeline binds as a dual orthosteric and allosteric ligand
Unambiguous cryo-EM density corresponding to xanomeline was observed in the highly conserved orthosteric site of the $M_4$ mAChR, as well as the common extracellular vestibule (ECV) allosteric site (Fig. 1a). In the orthosteric site, the thiadiazole-dihydropyridine core of xanomeline is positioned under a closed tyrosine lid consisting of residues Y113[3.33], Y416[6.51], and Y439[7.39] (superscript refers to the Ballesteros and Weinstein scheme for conserved Class A GPCR residues[24]) and above W413[6.48] in a position similar to iperoxo and acetylcholine (ACh) in prior mAChR structures (Fig. 1b–e). Specifically, the nitrogen atom from the dihydropyridine occupies a similar position as the nitrogen atom from the trimethyl-ammonium ion of iperoxo and ACh (Fig. 1c–e, g) allowing for an interaction with nearby residue D112[3.32], a conserved residue among aminergic GPCRs that typically interacts with the positively charged nitrogen atoms common in many orthosteric aminergic ligands[25]. Additional interactions between the core of xanomeline and the established mAChR orthosteric site[26,27–29] include residues N117[3.37], W164[4.57], N417[6.52], C442[7.42], Y443[7.43], and S116[3.36] (Fig. 1c). Previous $M_4$ mAChR mutagenesis studies support the interaction of xanomeline with these residues, as mutation of Y439[7.39], C442[7.42], and Y443[7.43] to alanine produced a significant decrease in xanomeline affinity[30].

In contrast to previous agonists in mAChR structures[10,19,20,31], xanomeline has a hydrophobic tail that extends out of the main orthosteric pocket in a vertical pose, relative to the thiadiazole-dihydropyridine core, and occupies a hydrophobic sub-pocket formed by transmembrane (TM) and extracellular loop (ECL) residues L190[ECL2], T196[5.39], T199[5.42], A203[5.46] and V420[6.55] (Fig. 1c–m). These sub-pocket residues (TM5/6/ECL2) are conserved across mAChR subtypes with exception of L190[ECL2], which is a F at the $M_2$ mAChR; a difference that was exploited in a structure-based manner to design $M_3$ vs $M_2$ mAChR selective antagonists[32]. In addition, we recently used MD simulations and pharmacology experiments to show that the "efficacy-driven" selectivity of xanomeline between the $M_4$ and $M_2$ mAChR subtypes was due to differences between L190[ECL2] at the $M_4$ mAChR and F181[ECL2] at the $M_2$ mAChR[33]. The orthosteric pose of xanomeline from the xano-$M_4$R-$G_{i1}$ structure supports our prior MD predictions and suggests that the TM5/6/ECL2 sub-pocket could potentially be exploited to design mAChR selective ligands against the $M_2$ mAChR—the structurally most closely related (to the $M_4$) off-target mAChR subtype that can cause dose-limiting clinical side effects[34].

Unexpectedly, strong cryo-EM density was also observed in the common mAChR extracellular allosteric binding site (Fig. 2a, b). Xanomeline was modelled into this density, revealing an extended binding pose where the hexyloxy tail extends out of the allosteric binding site towards TM1. Here the hexyloxy tail interacts with S436[7.36], while the nitrogen atom of the dihydropyridine core binds between F186[ECL2] and W435[7.35] forming potential cation-π interactions[35]. The thiadiazole-dihydropyridine makes additional interactions with Y89[2.61],

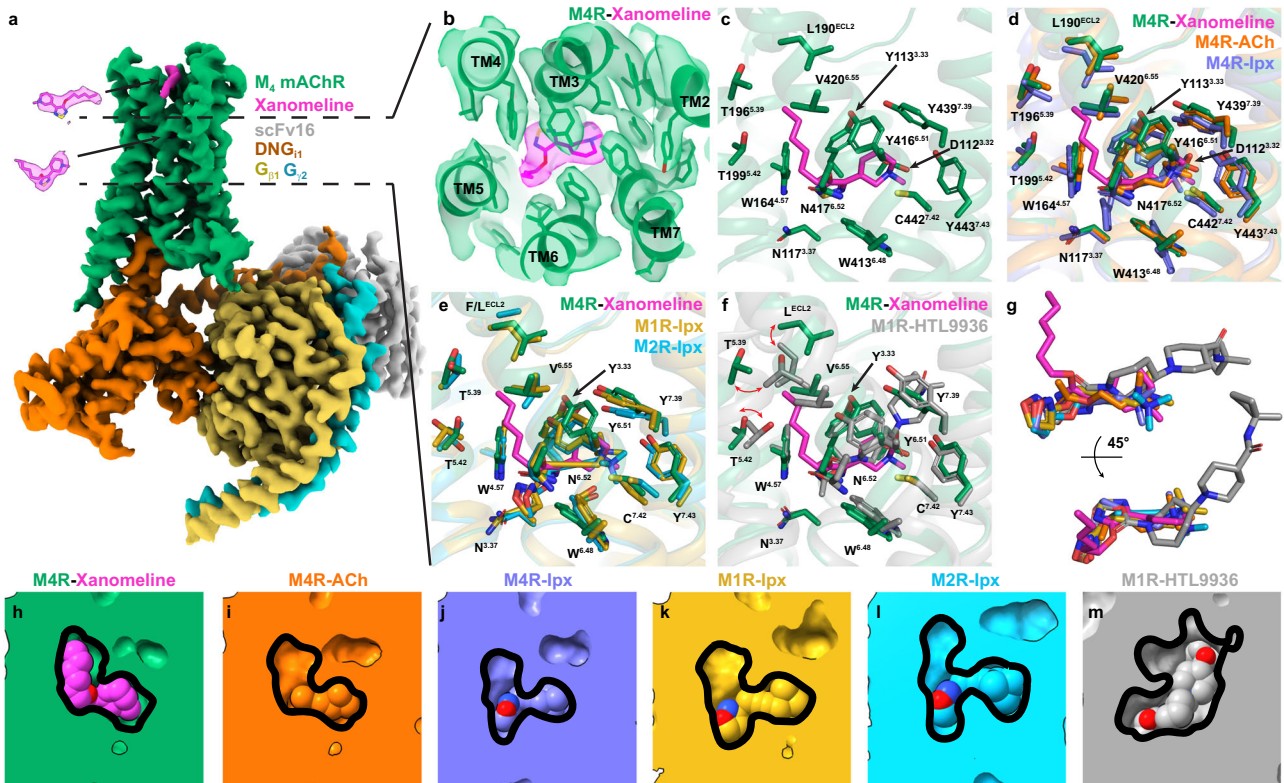

**Fig. 1 | Analysis of the orthosteric binding site of xanomeline. a** Consensus cryo-EM map of the $M_4$ mAChR (M4R) in complex with $DNG_{i1}/G\beta_1\gamma_2/scFv16$ bound to xanomeline resolved to 2.5 Å (FSC 0.143). The receptor is shown in green, the dominant negative (DN) heterotrimeric $G_{i1}$ protein is shown in orange, gold, and light blue for the α, β, γ subunits, respectively. Xanomeline is shown in magenta and scFv16 in silver. **b** Cryo-EM density (contour level 0.026) for xanomeline in the orthosteric binding site. **c** Xanomeline is bound in the canonical orthosteric binding site of the mAChRs positioned under a closed tyrosine lid composed of residues $Y^{3.33}$, $Y^{6.51}$ and $Y^{7.39}$. The hexyloxy tail of xanomeline sticks up towards the ECV region of the $M_4$ mAChR. **d** Comparison of the xanomeline bound active $M_4$ mAChR to the acetylcholine (ACh) and iperoxo (Ipx) bound $M_4$ mAChR (PDB: 7TRS and 7TRK, respectively). Orthosteric site residues of the xanomeline bound $M_4$ mAChR are shown as green sticks, residues of the ACh and Ipx bound $M_4$ mAChR are shown as orange and blue sticks, respectively. **e** Comparison of the xanomeline bound active

$M_4$ mAChR to the Ipx bound $M_1/M_2$ mAChRs (M1R/M2R, PDB: 6OIJ and 6OIK, respectively). Orthosteric site residues of the xanomeline bound $M_4$ mAChR are shown as green sticks, residues of the Ipx bound $M_1/M_2$ mAChRs are shown as yellow and light blue sticks, respectively. **f** Comparison of the xanomeline bound active $M_4$ mAChR to the HTL3396 bound $M_1$ mAChR (PDB: 6ZG4). Orthosteric site residues of the xanomeline bound $M_4$ mAChR are shown as green sticks, residues of HTL3396 bound $M_1$ mAChR are shown as grey sticks, respectively. **g** Overlay of xanomeline, ACh and Ipx bound to the $M_4$ mAChR, Ipx bound to the $M_1/M_2$ mAChRs and HTL9936 bound to bound $M_1$ mAChR. Cross section of the **h** xanomeline bound $M_4$ mAChR orthosteric binding site, **i** ACh bound orthosteric $M_4$ mAChR binding site, **j** Ipx bound $M_4$ mAChR orthosteric binding site, **k** Ipx bound $M_1$ mAChR orthosteric binding site, **l** Ipx bound $M_2$ mAChR orthosteric binding site, **m** HTL3396 bound $M_1$ mAChR orthosteric binding site.

$Y92^{2.64}$ and $D432^{7.32}$ (Fig. 2c). Comparing the allosteric binding pose of xanomeline to the binding pose of the PAMs, LY2033298 and VU046715, from our recent $M_4$ mAChR structures[20], reveals that the thiadiazole group of xanomeline overlaps with the carboxamide group of LY2033298 and VU0467154, while the tetrahydropyridine group of xanomeline is positioned just above the thieno[2,3-b]pyridine core of LY2033298 and VU0467154 (Fig. 2d, f). The hexyloxy tail of xanomeline occupies a similar position as the sulfonyl group of VU0467154. Similarly, xanomeline's thiadiazole group is positioned above the thieno[2,3-b]pyridine core of LY2119620 at the LY2119620-ipx-bound $M_2$ mAChR structure (Fig. 2e, f)[19,31].

Comparison of residues in the allosteric site of the $M_2$ and $M_4$ mAChR structures, each co-bound to an agonist and a PAM, with the xano-$M_4$R-$G_{i1}$ structure revealed that most allosteric site residues occupy a similar position and conformation (Fig. 2d–f). Importantly, the conformation of $W435^{7.35}$ was in a vertical pose, in line with other PAM-agonist-bound mAChR structures, compared to a horizontal pose observed in the agonist-bound mAChR structures[19,20,31]. Nevertheless, given the unexpected nature of this finding, we sought to further validate the allosteric nature of xanomeline through molecular dynamics (MD) simulations and molecular pharmacology experiments.

## Validation of the allosteric binding mode of xanomeline

To corroborate the interaction of xanomeline with the allosteric site of the $M_4$ mAChR, we utilized all-atom MD simulations and pharmacological validation. First, we initiated simulations of the $M_4$ mAChR with xanomeline bound in the orthosteric site, no ligand in the allosteric site, and xanomeline in solution (5 independent simulations, each 2 μs long) (Fig. 3a). Xanomeline bound spontaneously to the allosteric site in each of the 5 simulations, often staying bound for the remainder of the simulation. The xanomeline molecule at the orthosteric site also remained bound. For comparison, we initiated similar simulations with either the well-characterized PAM, LY2033298, or the orthosteric agonist, iperoxo, replacing xanomeline in solution. These simulations showed that the binding dynamics of xanomeline in the allosteric site closely resemble those of LY2033298. Xanomeline and LY2033298 bound to the allosteric site for a similar fraction of simulation time ($72 \pm 8\%$ and $64 \pm 17\%$, respectively). In contrast, the orthosteric agonist, iperoxo, interacted with the allosteric binding site only transiently and for a much lower fraction of simulation time ($12 \pm 5\%$), suggesting that it cannot interact in a stable manner with the allosteric site compared to both LY2033298 and xanomeline (Fig. 3a). Several iperoxo-bound mAChR structures support this finding, as these show iperoxo bound in the orthosteric site only[19,20,22,31,36]. It is possible that iperoxo

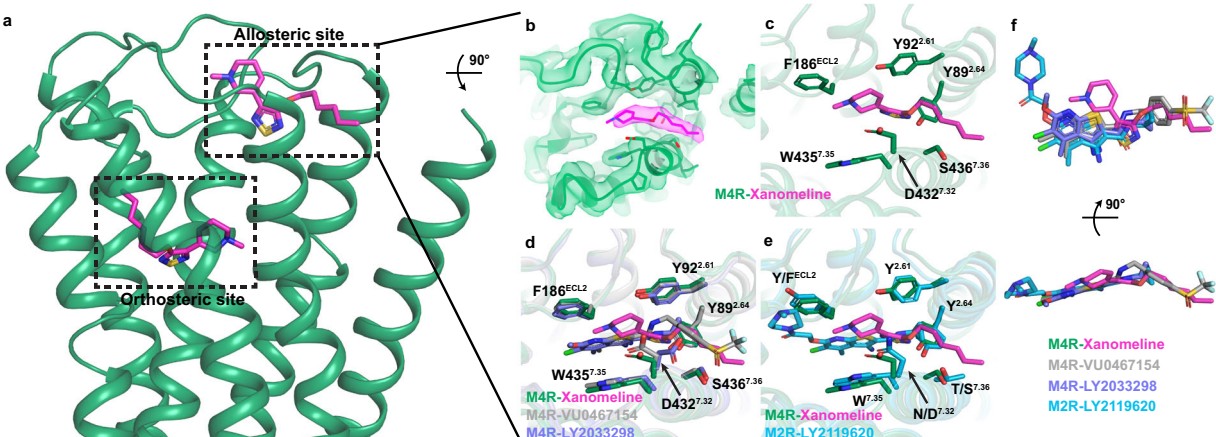

**Fig. 2 | Analysis of the allosteric binding site of xanomeline. a** Xanomeline is bound in both the orthosteric and allosteric binding sites of the M$_4$ mAChR (M4R). **b** Cryo-EM density (contour level 0.026) for xanomeline in the allosteric binding site. **c** Xanomeline in the common 'ECV' mAChR allosteric binding site with allosteric site residues shown as sticks in green. **d** Comparison of the xanomeline allosteric binding site to LY2033298 and VU0467154 bound to the allosteric binding site of the M$_4$ mAChR (PDB: 7TRP and 7TRQ). The allosteric binding site residues of the xanomeline bound M$_4$ mAChR are shown as green sticks whereas the allosteric binding site residues of the LY2033298 and VU0467154 bound M$_4$ mAChR are shown in blue and grey sticks, respectively. **e** Comparison of the xanomeline allosteric binding site to LY2119620 bound to the allosteric binding site of the M$_2$ mAChR (M2R, PDB: 6OIK). The allosteric binding site residues of the M$_4$ mAChR are shown as green sticks whereas the allosteric binding site residues of the M$_2$ mAChR are shown in light blue sticks. **f** Overlay of M$_4$ mAChR bound xanomeline, LY2033298, VU0467154 and M$_2$ mAChR bound LY2119620.

does not stably interact with the allosteric site because it is smaller in size and lacks aromatics rings that are present in both LY2033298 and xanomeline. Our MD simulations support the structural finding that xanomeline, unlike iperoxo, binds stably to the allosteric site of the M$_4$ mAChR, in addition to its ability to also bind to the orthosteric site. Although xanomeline adopted several allosteric binding poses during simulation, the most common pose was similar to that observed in the cryo-EM structure, forming interactions with TM2 and TM7 (Supplementary Fig. 5a, b).

We next performed radioligand dissociation experiments with the orthosteric antagonist [$^3$H]-N-methylscopolamine ([$^3$H]-NMS) using isotopic dilution via a saturating (10 μM) concentration of the antagonist, atropine, in the absence or presence of varying concentrations of xanomeline (Fig. 3b). The addition of a saturating concentration of atropine prevents the rebinding of [$^3$H]-NMS to the orthosteric site, thus allowing for the determination of the [$^3$H]-NMS dissociation rate constant. If xanomeline has an allosteric effect on orthosteric ligand affinity, it would be expected to alter the dissociation rate of [$^3$H]-NMS that is pre-equilibrated at the orthosteric site of the M$_4$ mAChR[37,38]. The presence of any additional xanomeline binding to the orthosteric site would have no bearing on the results of these dissociation kinetic experiments, since only the rate of dissociation of pre-bound [$^3$H]-NMS is being measured, i.e., any alterations in its rate constant can only occur as a consequence of a conformational change in the receptor mediated via a spatially distinct binding site.

At the wild-type (WT) M$_4$ mAChR, increasing concentrations of xanomeline progressively slowed the dissociation of [$^3$H]-NMS in a similar manner to LY2033298, clearly indicating that both ligands are able to bind allosterically and change orthosteric ligand dissociation (Fig. 3b, Supplementary Table 2). In contrast, a saturating concentration of the orthosteric agonist, iperoxo, had no effect on [$^3$H]-NMS dissociation indicating that it could not allosterically modulate [$^3$H]-NMS binding. By plotting the observed [$^3$H]-NMS dissociation rate as a function of xanomeline concentration, we derived a pIC$_{50\text{-Diss}}$ value of 3.92 ± 0.03 (n = 12) for xanomeline binding to the allosteric site of the M$_4$ mAChR, i.e., the potency ('apparent' affinity) for the allosteric site of xanomeline for the [$^3$H]-NMS-occupied receptor is ~120 μM. To further corroborate an allosteric binding mode of xanomeline, as well as validate its concomitant orthosteric binding mode, we performed a novel variant of the [$^3$H]-NMS dissociation kinetic assay, using

xanomeline alone to initiate both isotopic dilution (manifested at the orthosteric site) as well as monitoring for concentration-dependent changes on radioligand dissociation (which would manifest via allosteric site binding). At a concentration of 10 μM, xanomeline prevented the rebinding of [$^3$H]-NMS to the same extent as 10 μM atropine (Supplementary Fig. 6), consistent with full occupancy of the orthosteric site by xanomeline. Lower concentrations of xanomeline, did not prevent the rebinding of [$^3$H]-NMS. Importantly, when the concentration of xanomeline was increased to 100 μM, [$^3$H]-NMS dissociation was slowed to the same extent as that observed in the presence of 10 μM atropine + 100 μM xanomeline (Supplementary Fig. 6). Collectively, our cryo-EM structure, MD simulations, and pharmacological kinetic binding assays provide three distinct, but complementary, lines of experimental evidence that xanomeline can concomitantly occupy both orthosteric and allosteric sites at the M$_4$ mAChR.

Within the common ECV allosteric site, F186$^{ECL2}$ facilitates a π stacking interaction with xanomeline, as was observed with other prototypical allosteric modulators and the receptor[19,20,31]. At the M$_4$ mAChR F186A$^{ECL2}$ mutant, xanomeline no longer retarded [$^3$H]-NMS dissociation, further supporting that the ECV, and residue F186$^{ECL2}$ in particular, are needed for the allosteric binding of xanomeline at the M$_4$ mAChR (Fig. 3b). As expected, a similar effect was observed for the well-studied PAM, LY2033298, at this mutant. A loss in xanomeline modulation was also observed at other key ECV allosteric binding site mutants Y92$^{2.64}$A, Q184$^{ECL2}$A, W435$^{7.35}$A (Supplementary Fig. 7, Supplementary Table 2). Interestingly, mutation of allosteric residue Y89$^{2.61}$A led to an improved ability of xanomeline to further slow [$^3$H]-NMS dissociation, however, the same effect was observed when LY2033298 was tested at this mutant. Therefore, our mutagenesis experiments further support the common ECV as the allosteric binding site for xanomeline, given that it responds in the same manner as LY2033298 to residue changes within this site.

To determine whether xanomeline acts allosterically at the other mAChR subtypes, we tested xanomeline in [$^3$H]-NMS dissociation experiments at the remaining mAChR subtypes (Fig. 4a–d). A statistically significant difference in the dissociation of [$^3$H]-NMS, in the presence of 100 μM xanomeline, was observed at all mAChR subtypes (Supplementary Table 2). Despite this, the allosteric effect appeared modest at the M$_3$ and M$_5$ mAChRs and, consequently, xanomeline displayed reduced potency at these subtypes (Supplementary Fig. 7a,

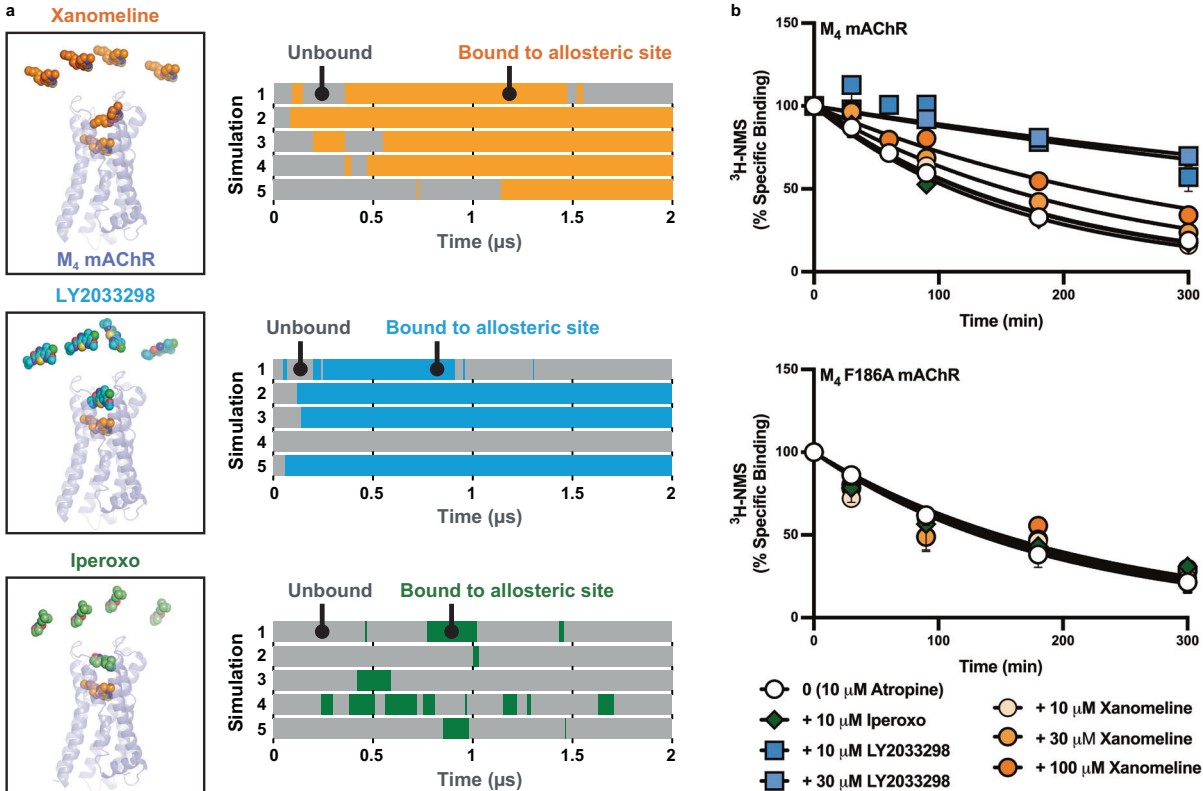

**Fig. 3 | Computational and pharmacological validation of xanomeline in the allosteric binding site. a** Molecular dynamics simulations reveal that xanomeline spontaneously binds to the $M_4$ mAChR allosteric site for a similar fraction of time as the prototypical PAM, LY2033298, and for a substantially longer fraction of time than the orthosteric agonist, iperoxo. Simulations were initiated with a xanomeline molecule bound in the orthosteric site and with the free ligands in solution—either xanomeline, LY2033298 or iperoxo—all being at the same concentration. Each horizontal bar represents an independent simulation and indicates the amount of time that the allosteric site is vacant (grey) or ligand-bound (non-grey). **b** [$^3$H]-N-methylscopolamine ([$^3$H]-NMS) dissociation via isotopic dilution with 10 μM atropine alone (0), or in the presence (+), of xanomeline, LY2033298, or iperoxo, at the $M_4$ mAChR wild type and $M_4$ F186[ECL2]A mutant. Data points represent the mean ± S.E.M. of three to nine individual experiments performed in duplicate. $M_4$ mAChR wild type; 10 μM atropine alone $n = 14$, + 10 μM iperoxo $n = 5$, + 30 μM LY2033298 $n = 7$, + 10 μM LY2033298 $n = 4$, + 10 μM xanomeline $n = 6$, + 30 μM xanomeline $n = 8$, + 100 μM xanomeline $n = 13$. $M_4$ F186[ECL2]A; 10 μM atropine alone $n = 4$, + 10 μM iperoxo & + 30 μM LY2033298 & + 30 μM xanomeline & + 100 μM xanomeline $n = 3$. A one-phase exponential decay model was fit to the data.

Supplementary Table 2). The allosteric effect was more pronounced at the $M_2$ mAChR and consequently, xanomeline had greater allosteric potency at this subtype. To further investigate xanomeline allosteric binding at other mAChR subtypes, we constructed several structural models. We placed xanomeline into the allosteric site of either an active-state structure ($M_1$, $M_2$ mAChRs) or an active-state homology model ($M_3$, $M_5$ mAChRs) and performed an energy minimization to determine whether the allosteric site can accommodate xanomeline binding at these subtypes (see Methods). For all subtypes, xanomeline and the surrounding residues displayed similar poses (Fig. 4e–i). Moreover, in MD simulations, xanomeline displayed a similar binding profile at all five mAChR subtypes (5 independent simulations per subtype, each 1 μs long) (Supplementary Fig. 5c). Interestingly, none of the residues in the allosteric site that contact xanomeline are conserved (Fig. 4j), which may explain the differences in the pharmacology. Further validation of xanomeline binding allosterically at other mAChR subtypes would be best validated by additional cryo-EM structures, which are beyond the scope of this study.

## Discussion

GPCRs are the largest class of medicinal drug targets, and recent discoveries have revealed a myriad of binding loci by which drug candidates interact with these receptors[39–42]. Here we report a cryo-EM structure of the clinically relevant, potential first-in-class, antipsychotic, xanomeline, bound to the human $M_4$ mAChR. The main finding from this work is that two molecules of xanomeline can bind

concomitantly to a single $M_4$ mAChR; one molecule within the classic orthosteric site, and a second molecule in an ECV allosteric binding site that has been well-characterized across all mAChR subtypes. To our knowledge, this is the first time that a small molecule clinical drug candidate has been shown to exhibit such concomitant orthosteric and allosteric binding modes on a single GPCR. As a result, our findings have a number of implications for our understanding of how GPCRs can be modulated, in addition to how structural biology can be used to inform drug discovery and understanding GPCR drug mechanisms.

At the orthosteric site of the $M_4$ mAChR, xanomeline bound in a pose that overlaps with the orthosteric agonists, iperoxo and ACh, but differs in that the hydrophobic tail of xanomeline extends into a sub-pocket derived from residues in TM5/6/ECL2. In a parallel study, we have shown that residue L190[ECL2] from the sub-pocket is conserved among mAChRs, with the exception of the $M_2$ mAChR where it is replaced by F181[ECL2]. Importantly, the difference in residues in this position is a major contributor to the "efficacy-driven" selectivity of xanomeline between the $M_4$ vs $M_2$ mAChRs[33]. Very recent structures of the $M_3$ and $M_4$ designer receptor exclusively activated by designer drugs (DREADD) mAChRs revealed that mutation of Y[3.33] at the DREADD creates a more open orthosteric binding pocket, such that DREADD selective ligands can extend into the TM5/6/ECL2 sub-pocket to achieve selectivity[36]. At the $M_1$ mAChR, structure-based drug design of the bulky agonist, HTL9936, revealed an orthosteric sub-pocket formed by residues in TM2 and TM3 (Fig. 1f, g) that facilitates subtype selectivity[10]. Similarly, the $M_2$ mAChR preferring orthosteric

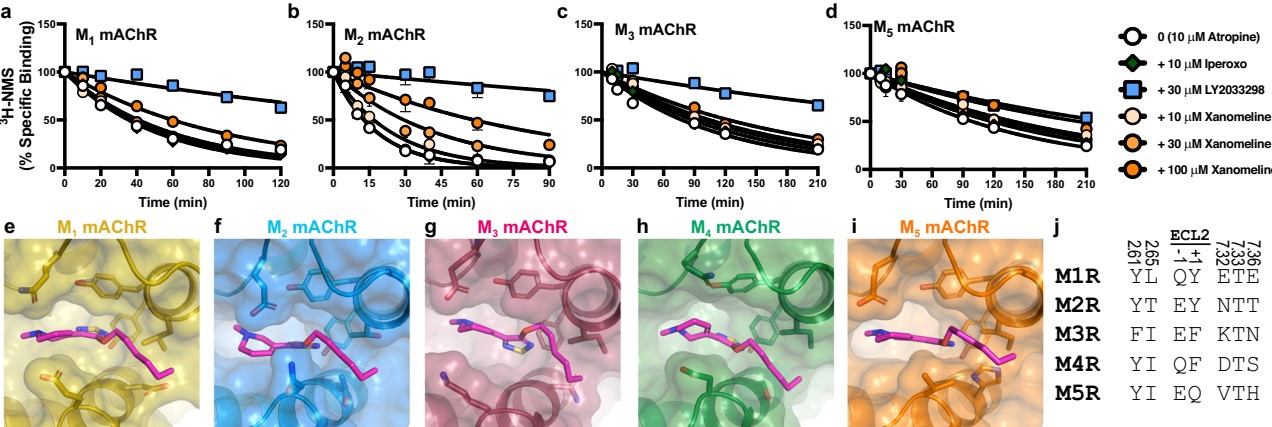

**Fig. 4 | Xanomeline binds allosterically at all mAChR subtypes. a–d** [³H]-N-methylscopolamine ([³H]-NMS) dissociation via isotopic dilution with 10 μM atropine alone (0), or in the presence (+), of xanomeline, LY2033298, or iperoxo, at the **a** M₁, **b** M₂, **c** M₃ or **d** M₅ mAChRs (M1R-M5R). Data points represent the mean ± S.E.M. of four to twelve individual experiments performed in duplicate. M₁ mAChR; 10 μM atropine alone & + 10 μM iperoxo & + 10 μM xanomeline & + 30 μM xanomeline & + 100 μM xanomeline $n = 6$, + 30 μM LY2033298 $n = 3$. M₂ mAChR; 10 μM atropine alone $n = 10$, + 10 μM iperoxo & + 30 μM LY2033298 & + 10 μM xanomeline & + 30 μM xanomeline & + 100 μM xanomeline $n = 6$. M₃ mAChR; 10 μM atropine alone & + 10 μM iperoxo & + 10 μM xanomeline & + 30 μM xanomeline & + 100 μM xanomeline $n = 7$, + 30 μM LY2033298 $n = 6$. M₅ mAChR; 10 μM atropine alone & +

10 μM iperoxo & + 10 μM xanomeline & + 30 μM xanomeline & + 100 μM xanomeline $n = 5$, + 30 μM LY2033298 $n = 4$. A one-phase exponential decay model was fit to the data. The allosteric binding site of each mAChR subtype can accommodate the binding of xanomeline, as shown by energy minimization of xanomeline in the allosteric site of the **e** M₁ (PDB: 6OIJ), **f** M₂ (PDB: 6OIK), **g** M₃, **h** M₄, or **i** M₅ mAChRs. **j** Sequence comparison of residues that contact xanomeline in the allosteric site (≤4 Å) across all five mAChR subtypes. Residues in TM2 and TM7 are labelled with the Ballesteros and Weinstein scheme for class A GPCRs and residues in ECL2 are numbered according to their relative position of a conserved cysteine residue (C185 at the M₄ mAChR).

antagonist, AF-DX 384, extends into a TM2/3 sub-pocket[43]. Collectively, these studies illustrate the potential for developing subtype selective mAChR ligands by targeting distinct orthosteric sub-pockets that exist in a conformation dependent on the orthosteric ligand. The identification of additional conformationally sensitive orthosteric sub-pockets may thus be a more general phenomenon than previously appreciated and can aid in selective orthosteric drug design but requires additional structural studies with novel ligands for further validation.

The finding that xanomeline bound to the ECV allosteric site was unexpected, because there is no prior evidence in the literature for xanomeline displaying unambiguous allosteric or cooperative behaviour in either binding or function[44]. However, this type of result is not totally without precedent. For example, prior pharmacological studies at the M₂ mAChR using the orthosteric antagonist, NMS, and the orthosteric agonist, oxotremorine-M, suggested that both ligands had potential to bind allosterically, albeit at extremely high concentrations with predicted affinities greater than 1 mM[45,46]. In contrast, the apparent affinity of xanomeline for the M₄ mAChR allosteric site from our study was substantially higher. Furthermore, the classical allosteric ternary complex model predicts that the potency (IC₅₀-Diss) of an allosteric modulator in retarding the dissociation rate of an orthosteric ligand is a function of both its affinity for the free allosteric site in the absence of co-bound orthosteric ligand ($K_B$) and the degree of cooperativity (α) between both orthosteric and allosteric sites when each site is occupied, specifically: IC₅₀-Diss = $K_B/α$. This relationship thus allows us to predict a potential affinity of xanomeline for the free allosteric site[47]. For instance, if the cooperativity between xanomeline and [³H]-NMS is neutral, i.e., α = 1, then the allosteric $K_B$ of xanomeline would remain 120 μM. In contrast, however, if the cooperativity between xanomeline and [³H]-NMS were negative, i.e., 0 < α < 1, then the affinity of xanomeline for the allosteric site would be substantially higher in the absence of orthosteric ligand binding. It has been shown that values of α less than 0.01 result in negative cooperativity that is virtually indistinguishable from a competitive interaction[48]. Given that xanomeline completely inhibits the equilibrium binding of [³H]-NMS, and assuming that this is (at least partly) a result of high negative

cooperativity, theoretical threshold α values ranging from 0.01–0.001 for this interaction would yield allosteric site $K_B$ estimates ranging from 1.2 μM–120 nM, respectively; similar to xanomeline's estimated orthosteric site affinity (158 nM), and thus possibly explaining why xanomeline displays a stable concomitant orthosteric and allosteric binding profile. Our structural findings also support the hypothesis that xanomeline has a binding affinity greater than 120 μM for the allosteric site. Specifically, high concentrations of ligands are routinely used in structural studies of GPCRs to ensure high ligand occupancy and receptor-complex stability without resulting in the detection of secondary ligand densities. In our study, we used 50 μM xanomeline during the purification of the xano-M4R-G_{i1} complex, which would represent less than 50% receptor occupancy if its affinity for the free allosteric site was only 120 μM. This would be highly unlikely to result in stable binding or clear resolution of ligand density at this site, in contrast to what was actually observed. Therefore, we do not believe that the allosterically-bound xanomeline was an artifact due to the high concentration used for the cryo-EM structure, but rather a pharmacologically-relevant phenomenon for the reasons outlined above. Moreover, our MD simulations, pharmacology and mutagenesis experiments also confirmed a true allosteric interaction.

Recent agonist-bound x-ray crystallography structures of the M₁ mAChR reveal lipid molecules binding near TM1 and the ECV allosteric site, where they would reside in close position to the hexyloxy tail of xanomeline (Supplementary Fig. 8)[10]. In fact, MD simulations at several GPCRs, including the M₂ and M₃ mAChRs, have predicted that, in order to enter the orthosteric site, many orthosteric ligands may need to form a 'metastable' intermediate conformation in the extracellular region of the receptor[49–51]. The ECL regions of GPCRs can impact the dissociation rate of orthosteric ligands supporting the potential for extracellular allosteric sites to influence these intermediate conformations[28,52,53]. These prior observations are not mutually exclusive with our current findings. Rather, our study highlights the potential that some orthosteric ligands may possess concomitant allosteric properties if their 'metastable' interactions are of sufficient affinity, and thus more stable, than previously appreciated.

The allosteric binding of xanomeline may also explain the confounding interaction previously characterised between xanomeline and the PAM, LY2033298, at the $M_2$ mAChR[54]. In contrast to strong positive cooperativity observed for the interaction between LY2033298 and the high efficacy agonist, oxotremorine-M, at the $M_2$ mAChR, the interaction between LY2033298 and xanomeline at this mAChR subtype was characterised by modest positive cooperativity at the level of binding affinity, but negative cooperativity at the level of $M_2$ mAChR function, i.e., xanomeline's efficacy was abolished upon increasing concentrations of LY2033298[54,55]. At the time, this unusual finding was interpreted as an example of 'probe-dependence', whereby the interaction between LY2033298 and xanomeline produced a unique, non-signalling, conformation of the $M_2$ mAChR. In addition, a previous study of the interaction between xanomeline and LY2033298 at the mouse $M_4$ mAChR revealed a weak positively cooperative effect, which was interpreted in terms of potential species variability between human and rodent mAChRs[56]. However, in light of our new structural studies and demonstration that xanomeline displays allosteric binding properties at the $M_2$ mAChR in addition to the $M_4$ mAChR, it is now possible that both of these prior findings may reflect, at least in part, an unappreciated competitive interaction between LY2033298 and xanomeline for the ECV allosteric site at both receptors.

Collectively, our results shed new light on the actions of a clinical drug candidate for the treatment of schizophrenia. The identification and characterisation of both orthosteric and allosteric binding modes is a clear point of differentiation to previous GPCR structures and may help explain prior aspects of the atypical pharmacology of xanomeline. The extent to which this novel mode of 'dual-site, single-target' concomitant engagement translates to other GPCRs remains to be determined. Nonetheless, the demonstration that a late-stage clinical GPCR drug candidate can engage both orthosteric and allosteric sites represents a novel finding that further highlights the myriad of mechanisms by which GPCRs can be regulated, and serves as in impetus for further studies on other drug candidates for this important protein family.

## Methods

### Expression and purification of scFv16

A C-terminal 8xhistidine-tagged single chain construct of Fab16 (scFv16) was cloned into a modified pVL1392 baculovirus transfer vector and expressed in secreted form using the BestBac Baculovirus Expression System (Expression Systems) in *Trichoplusia ni* (Tni) insect cells. Cells were grown in ESF 921 serum-free media (Expression System) and infected at a density of $4.0 \times 10^6$ cells per mL and shaken at 27 °C for 48–72 h. To purify scFv16, supernatant from baculovirus-infected cells was pH balanced with Tris pH 8.0. Chelating agents were quenched by addition of 5 mM $CaCl_2$ and incubation with stirring for 1 h at 25 °C. Resulting precipitates were removed by centrifugation and the supernatant was loaded onto Ni-NTA resin. The column was washed with 20 mM HEPES pH 7.5, 50 mM NaCl, and 10 mM imidazole followed by a low salt wash comprised of the same buffer substituted with 100 mM NaCl. Protein was eluted with low salt wash buffer supplemented with 250 mM imidazole. SDS-page with Coomassie staining was performed on the elution to determine purity. Sample was concentrated, flash frozen using liquid nitrogen and stored at −80 °C.

### Expression and purification of Xano-M$_4$ $_{ΔICL3}$–dnG$_{i1}$ complex

A diagram of the expression and purification of Xano-M$_4$ $_{ΔICL3}$–dnG$_{i1}$ complex is provided in Supplementary Fig. 1. The human $M_4$ muscarinic receptor gene was modified to contain an N-terminal Flag tag and a C-terminal 8×histidine tag. In order to increase stability and expression, residues 242–387 of intracellular loop 3 (ICL3) were removed. In addition, the N-terminal glycosylation sites (N3, N9, N13) were mutated to Asp. The resulting Flag-M$_{4ΔICL3}$-His construct was

cloned into a pVL1392 baculovirus transfer vector and expressed in *Spodoptera frugiperda* (Sf9) cells. Human dominant negative (DN) DNGα$_{i1}$ and His6-tagged human Gβ$_1$γ$_2$ were co-expressed in Tni insect cells. Sf9 and Tni cells were grown in ESF 921 serum-free media (Expression System) and infected with either $M_4$ mAChR virus or a 1:1 ratio of dnG$_{i1}$ and Gβ$_1$γ$_2$ virus, respectively, at a density of $4.0 \times 10^6$ cells per mL. $M_4$ mAChR expression was supplemented with 10 μM atropine (Atr). Cells were shaken at 27 °C for 48–60 h and then harvested by centrifugation (10,000 × g, 20 min, 4 °C). Cell pellets were flash frozen using liquid nitrogen and stored at −80 °C. To begin complex purification, $M_4$ mAChR was solubilised in 20 mM HEPES pH 7.5, 10% glycerol, 750 mM NaCl, 5 mM $MgCl_2$, 5 mM $CaCl_2$, 0.5% lauryl maltose neopentyl glycol (LMNG), 0.02% cholesteryl hemisuccinate (CHS), 10 μM Atr and supplemented with complete Protease Inhibitor Cocktail tables (Roche) and stirred at 4 °C for 2 h. Sample was centrifuged (30,000 × g, 30 min, 4 °C) and supernatant was filtered and batch bound to M1 anti-flag affinity resin for 1 h at 25 °C. Resin was loaded onto glass column and washed with wash buffer A (20 mM HEPES pH 7.5, 750 mM NaCl, 5 mM $MgCl_2$, 5 mM $CaCl_2$, 0.5% LMNG and 0.02% CHS) for 30 min at 2 mL/min and with wash buffer B (20 mM HEPES pH 7.5, 100 mM NaCl, 5 mM $MgCl_2$, 5 mM $CaCl_2$, 0.5% LMNG, 0.02% CHS and 50 μM xanomeline). At the same time DNGα$_{i1}$β$_1$γ$_2$ pellet was solubilised in 20 mM HEPES pH 7.5, 100 mM NaCl, 5 mM $MgCl_2$, 5 mM $CaCl_2$, 0.5% LMNG, 0.02% CHS, apyrase and supplemented with complete Protease Inhibitor Cocktail tables (Roche). Sample was incubated at 4 °C for 2 h. The solubilised G protein was then added to receptor bound to M1 anti-flag affinity resin and supplemented with apyrase, 50 μM xanomeline and scFv16 and incubated at room temperature for 1 h. Following incubation, resin was packed into glass column, washed with 20 mM HEPES pH 7.4, 100 mM NaCl, 5 mM $MgCl_2$, 5 mM $CaCl_2$, agonist, 0.01% LMNG and 0.001% CHS. Sample was eluted with SEC buffer (20 mM HEPES pH 7.4, 100 mM NaCl, 5 mM $MgCl_2$, 50 μM xanomeline, 0.005% LMNG and 0.0005% CHS) supplemented with 10 mM EDTA and 0.1 mg/mL Flag peptide. Elution was concentrated and run through SEC using a Superdex200 increase 10/300 column (Cytiva) with SEC buffer. Fractions contained sample were pooled, concentrated to 3–5 mg/mL, flash frozen using liquid nitrogen and stored at −80 °C.

### EM sample preparation and data acquisition

3 μL of sample was applied to glow-discharged UltrAuFoil R1.2/1.3 Au 300 mesh grids (Quantifoil) and vitrified on a Vitrobot Mark IV (Thermo Fisher Scientific) set to 4 °C and 100% humidity and 10 s blot time. Data were collected on a Titan Krios G3i 300 kV electron microscope (Themo Fisher Scientific) equipped with GIF Quantum energy filter and K3 detector (Gatan). Data acquisition was performed in EFTEM NanoProbe mode with a 50 μM C2 aperture at an indicated magnification of ×105,000 with zero-loss slit width of 25 eV. The data was collected automatically with homemade scripts for SerialEM[57] performing a 9-hole beam-image shift acquisition scheme with one exposure in the centre of each hole. Experimental parameters are listed in Supplementary Table 1.

### Image processing

Specific details for the processing of the Xano-M$_4$ $_{ΔICL3}$–dnG$_{i1}$ complex cryo-EM data set are shown in Supplementary Fig. 2. 5707 micrographs were motion corrected through UCSF MotionCorr[58] and contrast transfer function (CTF) estimated through CTFFIND 4.1.8[59]. Using the corrected micrographs, particles were picked using the automated template picking routine in RELION3.1[60] and these were used in reference free 2D and 3D classification. Particles contributing to bad classes were removed and particles contributing to good classes were subjected to further analysis and processing in Bayesian polishing, CTF refinement and 3D auto-refinement followed by another round of 3D classification and 3D refinement in RELION that yielded the final

maps[60]. Local resolution was determined from RELION using half-reconstructions as input maps.

## Model building and refinement

The active structure of $M_4R$ bound to $G_{i1}$ and iperoxo ($M_4R$-$DNG_{i1}$-ipx, PDB Code: 7TRK)[20] was used as an initial template for model building. The ligand was removed from the model before receptor, G protein and ScFv16 were rigid body placed in the EM map using UCSF ChimeraX v1.15[61]. The model was refined using repeated rounds of manual model building in Coot v0.9[62] and real space refinement in Phenix v1.20.1[63]. Ligand restraints were generated using the GRADE server (http://grade.globalphasing.org). Model quality was assessed using MolProbity v4[64] and the wwPDB validation server before deposition in the wwPDB[65] (PDB: 8FX5 and EMDB-29524). Structure figures were generated using UCSF ChimeraX v1.15[61] and PyMOL v2.5 (Schrödinger). The active state structures of $M_3$ and $M_5$ receptors (Fig. 4) were prepared using homology modelling in Prime v2020-1 (Schrodinger) using the $M_2$ structure (PDB: 6OIK) as a template. To construct models of xanomeline allosterically bound to the other mAChR subtypes (Fig. 4), we first attempted to perform ligand docking. However, docking produced many possible binding poses with similar energy scores. It could not reproduce the allosteric xanomeline binding pose observed in the $M_4$ cryo-EM structure as the most favourable pose. This may be because docking does not take into account the membrane interface near xanomeline's tail. Therefore, we placed xanomeline in the allosteric site and performed energy minimization using the OPLS forcefield in the Schrodinger software suite.

## Mammalian cell culture

Flp-In Chinese hamster ovary (CHO) (Thermo Fisher Scientific) cells stably expressing mAChR WTs or mutant constructs were cultured in Dulbecco's modified Eagle's medium (DMEM, Invitrogen) containing 5% foetal bovine serum (FBS; ThermoTrace) and 0.6 μg/mL of hygromycin (Roche) in a humidified incubator (37 °C, 5% $CO_2$). Upon reaching confluence, media was removed, cells were washed with phosphate buffered saline (PBS) and harvested from tissue culture flasks using versene (PBS with 0.2 g EDTA). Cells were then pelleted through centrifugation at $350 \times g$ for three minutes followed by resuspension in DMEM + 5% FBS. Cells were then either plated for an assay or reseeded into a tissue culture flask.

## Radioligand dissociation binding

FlpIn CHO cells stably expressing mAChR WTs or mutants were plated at 20,000–50,000 cells per well in 96-well isoplates (PerkinElmer Life Sciences) and incubated overnight at 37 °C, 5% $CO_2$. The following day, cells were washed with 100 μL PBS and incubated in 1xHBSS binding buffer (138 mM NaCl, 5.3 mM KCl, 0.5 mM $MgCl_2$, 0.4 mM $MgSO_4$, 0.4 mM $KH_2PO_4$, 1.3 mM $CaCl_2$, 5.5 mM D-glucose, 0.3 mM $Na_2HPO_4$, 10 mM HEPES, pH 7.4) with a $K_D$ concentration of [³H]-NMS for a minimum of 3 h at room temperature in a total volume of 90 μL. Dissociation of the radioligand was initiated by addition of 10 μM Atropine alone or in the presence of different concentrations of xanomeline, 30 μM or 10 μM LY2033298 or 10 μM iperoxo at various timepoints. The assay was terminated through the rapid removal of the radioligand followed by two 100 μL washes of ice-cold 0.9% NaCl buffer. Radioactivity was determined by the addition of 100 μL of Optiphase Supermix scintillation fluid and counting in a MicroBeta2 plate counter (PerkinElmer Life Sciences).

## Data analysis

All data were analysed using GraphPad Prism 9 (Graphpad Software, San Diego, CA). Dissociation kinetic data were globally fitted to one-phase exponential decay function to derive the apparent rate constant of dissociation ($k_{off}$) in the absence or presence of each compound. Results are expressed as means and S.E.M. unless otherwise stated.

## Molecular dynamics

We performed simulations of the $M_4$ mAChR with xanomeline bound to the orthosteric site and either xanomeline, LY2033298, or iperoxo in solution. Simulations were initiated from the coordinates of the cryo-EM structure. The G protein and allosteric xanomeline molecule were removed. Six molecules of the ligand under consideration were placed in the extracellular region at least 20 Å away from the receptor. For all simulations, hydrogen atoms were added, and protein chain termini were capped with neutral acetyl and methylamide groups. Titratable residues were kept in their dominant protonation state at pH 7, except for $D^{2.50}$ and $D^{3.49}$, which were protonated (neutral), as studies indicate that these conserved residues are protonated in active-state GPCRs[66,67]. Histidine residues were modelled as neutral, with a hydrogen atom bound to epsilon nitrogen. The amine of xanomeline was protonated to form a salt bridge with the conserved aspartate in the orthosteric binding site, in alignment with other muscarinic agonists. The Dowser programme was used to hydrate pockets within and around each structure[68]. Then the receptor was inserted into a pre-equilibrated palmitoyl-oleoyl-phosphatidylcholine (POPC) bilayer using Dabble (1999 release)[69]. Sodium and chloride ions were added to neutralize each system at a concentration of 150 mM. The final systems comprised ~62,000 atoms, including ~130 lipid molecules and ~13,300 water molecules. Approximate system dimensions were $80 \text{ Å} \times 80 \text{ Å} \times 100 \text{ Å}$.

All simulations were run on a single Graphics Processing Unit (GPU) using the Amber18 Compute Unified Device Architecture (CUDA) version of particle-mesh Ewald molecular dynamics (PMEMD)[70]. We used the CHARMM36m parameter set for protein molecules, lipids, and ions, and the CHARMM TIP3P water model for waters[71,72]. Parameters for ligands were generated using the CHARMM General Force Field (CGenFF) with the ParamChem server[73], Gaussian, and AmberTools Paramfit[74]. Minimization, heating, and equilibration steps were run as described recently[75]. Trajectory snapshots were saved every 200 ps. All simulations were at least 2 microseconds in length. The AmberTools18 CPPTRAJ package[76] was used to reimage trajectories, while Visual Molecular Dynamics (VMD)[77] and PyMol (Schrodinger) were used for visualization and analysis.

In Fig. 3a, the bound state was classified by measuring the minimum distance between the M4 mAChR allosteric site residues and the ligand molecules initially placed in solution. The allosteric site residues were defined as residues 184 to 190, 416, 419, 423, 439, 435, 432, 436, 440, 89, 92, 93, and 35 from inspection of the structure. The distance was smoothed over time by a uniform filter (moving average) with width of 30 ns to remove very short events. We chose a threshold distance of 3 Å between ligand and allosteric site residues to define the bound state. Each bar shows when any ligand is bound to the allosteric site over the course of the simulation time, including both initial equilibration and production frames.

## Reporting summary

Further information on research design is available in the Nature Portfolio Reporting Summary linked to this article.

## Data availability

The data that support this study are available from the corresponding authors upon request. Atomic coordinates and cryo-EM maps for the reported structures were deposited in the Protein Data Bank under accession code 8FX5 (Human M4 muscarinic acetylcholine receptor complex with Gi1 and xanomeline) and in the Electron Microscopy Data Bank under accession code EMD-29524 (Human M4 muscarinic acetylcholine receptor complex with Gi1 and xanomeline). Previously published structures can be accessed via accession codes: 7TRS, 7TRK, 7TRP, 7TRQ, 6OIJ, 6OIK, 6ZG4, 6ZFZ, 6ZG9, and 7V68. Simulation trajectories are available at https://doi.org/10.5281/zenodo.8136971.

The source data underlying Figs. 3 and 4 and Supplementary Figs. 6 and 7 are provided as a Source data file. The initial coordinate file, simulation input files, and trajectories are available on Zenodo (https://doi.org/10.5281/zenodo.8136971). Source data are provided with this paper.

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

## Acknowledgements

This work was funded with support by a Wellcome Trust Collaborative Award (201529/Z/16/Z: P.M.S., A.B.T., A.C.), the National Health and Medical Research Council of Australia (1055134: A.C., P.M.S.; 1150083: P.M.S., A.C.; and 1138448: D.M.T.), the Australian Research Council (DE170100152: D.M.T.; DP190102950: C.V., A.C., and IC200100052: P.M.S., D.W.), the National Institutes of Health (R01GM127359: R.O.D.), the National Science Foundation Graduate Research Fellowship (A.S.P.), and Karuna Therapeutics. P.M.S. is a Senior Principal Research Fellow (1154434), D.W. a Senior Research Fellow (1155302), D.M.T. an Early Career Research Fellow (1196951). R.D. was supported by Japan Society for the Promotion of Science (JSPS) Kakenhi grant (22H02554). This work was partially supported by the Monash University Ramaciotti Centre for cryo-electron microscopy and the Monash University MASSIVE high-performance computing facility and supercomputing resources.

## Author contributions

D.M.T., C.V., A.C., R.O.D. and A.B.T. designed the overall research. D.M.T. and Z.V. designed, expressed, and purified protein samples. A.G., Z.V. performed negative-stain EM. R.D. performed sample vitrification and cryo-EM imaging. D.M.T., J.I.M. and A.G. processed the EM data. D.M.T., J.I.M. and A.G. generated and analysed atomic models. A.S.P., Y.L. and R.O.D. designed, performed, and analysed MD simulations. W.A.C.B. and V.P. generated DNA constructs and performed pharmacology experiments. W.A.C.B., V.P., C.V., A.C. and D.M.T. analysed pharmacology data. D.W., P.M.S., A.C., C.V., D.M.T., R.O.D., A.B.T., C.C.F. and S.M.P. provided supervision. W.A.C.B., A.C. and D.M.T. wrote the manuscript with contributions and input from all authors.

## Competing interests

C.C.F. and S.M.P. are employees of and hold equity in Karuna Therapeutics. C.V., D.M.T. and A.C. have, and R.O.D. had, sponsored research agreements with Karuna Therapeutics. P.M.S. and A.C. are co-founders and hold equity in Septerna Inc. D.W. and R.O.D. hold equity in Septerna Inc. The remaining authors declare no competing interests.
