## [Peer Review File · Nature Communications]

Xanomeline displays concomitant orthosteric and allosteric binding modes at the M4 mAChRReviewers' Comments:

Reviewer #1:

Remarks to the Author:

In this manuscript, Burger et al. present evidence that the muscarinic acetylcholine receptor (mAChR) agonist xanomeline simultaneously binds to two sites on the M4 mAChR—the orthosteric binding site (as previously known) and a conserved extracellular allosteric binding site (where several other mAChR allosteric modulators are known to bind)—based on a cryo electron microscopy structure of xanomeline-M4 mAChR-heterotrimeric Gi. This finding was supported by (1) molecular dynamics simulations indicating that xanomeline spontaneously binds to the allosteric site and (2) radioligand binding experiments showing that xanomeline slows dissociation of the orthosteric antagonist [³H]-NMS from M4 and other mAChRs (thus demonstrating that xanomeline must be able to occupy the allosteric site). The authors suggest that this dual binding mode could be related to some of the distinctive pharmacological properties of xanomeline (e.g., its “wash-resistant” pharmacology and negative cooperativity with LY2033298, a positive allosteric modulator of other mAChR agonists).

The proposed mechanism is quite interesting and could be important for future drug design efforts for mAChR agonists. While interactions of mAChR orthosteric agonists with the allosteric site during the binding process have been previously proposed, this manuscript provides evidence that these interactions can be quite stable for some ligands. My main concern is the limited amount of experimental support for the allosteric binding mode and the absence of data to show that:

(1) xanomeline can in fact simultaneously bind to BOTH the orthosteric and allosteric sites. This could be assessed with radiolabeled xanomeline (if available) based on its binding stoichiometry. If not, since there are radiolabeled allosteric ligands commercially available (e.g., [³H]-LY2119620), it should at least be possible to demonstrate that xanomeline directly competes with the allosteric ligand.

(2) binding of xanomeline to the allosteric site contributes to its pharmacology. It is obviously challenging to assess the contribution of allosterically bound xanomeline to signaling since it is also an orthosteric agonist. However, the authors mention several phenomena that appear to be testable. For example, does the F186A mutation in the allosteric site affect xanomeline’s “wash-resistance”?

While I do not want to burden the authors with a number of additional experiments, there is currently only one set of experiments to support the dual orthosteric/allosteric binding mode (the central claim of the paper), so I think at least one more line of evidence to solidify this claim or demonstrate its pharmacological relevance is important.

Minor points:

1) Could you please clarify if the orthosteric xanomeline remains stably bound to this site throughout the duration of the MD simulation runs?

2) The authors and their collaborators have previously published MD simulations with xanomeline/M4 mAChR (e.g., ref 33). Could you please clarify whether stable binding to the allosteric site was observed in simulation there (and just not understood/commented upon), or whether simulation conditions were different in this manuscript that led to the stable allosteric site interactions observed?

3) The authors cite previous work in the discussion showing that LY2033298 and xanomeline show negative cooperativity. However, LY2033298 interacts stably with the allosteric site in simulation with xanomeline bound to the orthosteric site (Figure 3A). The authors may want to add a statement in the discussion to clarify explicitly why these findings are not incompatible.

Reviewer #2:

Remarks to the Author:

In this manuscript, Burger et al. report the active state cryo-EM structure of the human M4 mAChR bound to xanomeline and in complex with heterotrimeric Gi. Remarkably, the structure revealed two molecules of xanomeline bound to the receptor, one in the canonical orthosteric binding site and a second one in an extracellular vestibular allosteric site (that had been previously discovered in all muscarinic acetylcholine receptors). This novel binding mode is then investigated using molecular dynamics simulations and radioligand dissociation assays in receptor mutants.

This work provides yet another fascinating example of the wide diversity in binding modes displayed by GPCR ligands. Transient recognition by an extracellular 'vestibule' prior to orthosteric binding is an idea that has been going around the field for some time. While such allosteric vestibules are now well documented, this manuscript provides the first structural evidence of simultaneous binding of an agonist to both sites. The data presented are convincing and properly analyzed. The findings are original and are presented rigorously and clearly. The computational and experimental methods are adequate. In my opinion, this work is remarkable and interesting. I particularly appreciate the thought put on the discussion about the possible affinity of the allosteric binding site.

Still, there are some specific points in the manuscript that I think need further clarification, particularly regarding the molecular dynamics simulations and their interpretation.

Major comments

- page 3: "The percent protonation of the dihydropyridine nitrogen atom was predicated to be approx. 47% at pH 7.4". I could not find in the manuscript how this percentage was calculated. In any case, according to these numbers, half of the ligand molecules would be neutral and presumably have a low affinity for the orthosteric site. But would this affect binding to the allosteric site? I presume this would have consequences on the pharmacological properties of xanomeline. Can the authors elaborate on this? Also, I presume the authors mean 'predicted', not 'predicated'.

- page 5: "Xanomeline bound spontaneously to the allosteric site in each of the 5 simulations ..." I am surprised that the authors could observe spontaneous ligand binding in the scale of microseconds in classical molecular dynamics simulations. Usually, enhanced sampling methods are required to reproduce such binding events, which are usually much slower. Can the authors elaborate on this point? Is there anything special about the M4 mAChR or xanomeline/LY2033298 that suggest unusually fast on-rates? Have these values been experimentally measured?

- page 5: "... the orthosteric agonist, iperexo, interacted with the allosteric binding site only transiently... Xanomeline, unlike iperexo, binds stably to the allosteric site of the M4 mAChR..." Have the authors discussed what could be the cause of this difference? I could not find such analysis in the manuscript. Did I miss it?

- page 6: "... whereas an increase in the ability of xanomeline to modulate 3H-NMS dissociation was observed at Y89(2.61)A, ...". What does 'an increase in the ability... to modulate' mean? That the dissociation of xanomeline is slower in Y89(2.61)A than in other mutants? Do the authors have an hypothesis about the origin of this effect?

- page 6: "... and performed an energy minimization ...". Geometry optimization of the receptor-ligand complex by energy minimization only leads to a local minimum very near the starting pose. It is not clear to me that such strategy would necessarily accommodate xanomeline in the binding pocket of receptors of different subtypes in a native binding pose. Why didn't the authors start with computational docking experiments?

- page 8: "... but is most likely due to xanomeline interacting with lipids through its hexyloxy tail". In this paragraph, the authors discuss the possible binding mechanism of xanomeline and cite molecular dynamics simulation studies suggesting the existence of a channel in the transmembrane bundle to access the orthosteric binding site, or a three-step binding mode, in which xanomeline would interact with the lipid bilayer, diffuse into the allosteric site, and then move to the orthosteric site. I presume that none of this was observed in the simulations performed in this work, otherwise the authors would have discussed it. If this is the case, how was xanomeline binding to the receptor in their simulations? Directly diffusing from the solvent? Is there anything remarkable in the desolvation mechanism of the ligand? Were there any interactions of the ligand with the membrane at all? I think these are points that should be discussed in further detail.

Minor comments

- page 4: "Unexpectedly, strong cryo-EM density was also observed in the common mAChR extracellular allosteric binding site". Was this density observed in the maps of other major 'good' classes? Could the authors tell?

- page 7: "Collectively, these studies illustrate the potential for developing subtype selective mAChR ligands by targeting distinct orthosteric sub-pockets that exist in a conformationally dependent manner." I think I understand what the authors mean, but the expression 'exist in a conformationally dependent manner' is not very clear. The authors are referring to different conformations observed in 'static' structures, but the reader might think of 'dynamic conformations' (i.e. different conformations as a result of inherent protein dynamics). At least, I did in the first read. Can the authors please rephrase this argumentation?

- The manuscript as written is very interesting. However, I was expecting to read something about the interactions between the receptor and the G protein. As the authors did not say anything about it, I presume there is nothing remarkable in this interface. Did the authors look at it in any detail?

Reviewer #3:

Remarks to the Author:

In the manuscript by Wessel and collaborators, the authors describe the cryo-EM structure of active M4 muscarinic acetylcholine receptor (M4 mAChR) in complex with the G protein and agonist xanomeline. Besides the orthosteric (acetylcholine-binding) site, xanomeline was found to bind one extracellular vestibular site with the support of the solid density. To further validate this hypothesis, they performed molecular dynamics simulations and observed the binding of Xanomeline from solution to the extracellular allosteric site. Meanwhile, radioligand dissociation experiments of both wildtype and several mutants for M4 mAChR as well as other mAChRs were adopted to support the allosterically bind at all mAChR subtypes. The work provides the molecular pharmacology of xanomeline and expand the knowledge of GPCR ligand binding, which may enable the structure-based design of novel selective mAChR ligands.

The manuscripts are well-written and has shown remarkable novelty. Give there are multiple released structures of M4 mAChR, the novelty of current manuscript mainly focused on the "allosteric" binding site. However, current data is probably not suitable for a full manuscript (if possible, communication is perfect). Several major points as well as some minor points should be address before further consideration.

Major issues:

1. Clearly, the authors prefer describe the allosteric binding instead of the allosteric modulation, indeed as the data shown. The functional link between allosteric binding and allosteric modulation (mainly as downstream signaling) is not clear, i.e., how to design or optimizes these types of ligands toward better efficacy is very important and I guess several lessons could be obtain from this excellent special case. Are there any functional consequences of the F186A as well as Y92A, Q184A, W435A, and Y89A? How about their influence on endogenous agonist by xanomeline (antagonism for

orthosteric binding site, positive stabilization for allosteric site?)

2. As shown in Fig 3A, LY2033298 can bind to the allosteric site in a similar manner as xanomeline, while significant difference on radioligand dissociation experiments (Fig. 3B) was observed mainly due to the fact that xanomeline can also bind to the orthosteric site. Repeating these measurements with varying concentration of atropine may draw a clearer picture of cooperativity between orthosteric binding and allosteric binding. Meanwhile, a chemical structural comparison among these three ligands (LY2033298, iperoxo and xanomeline) should be included in Fig. 3.

3. The allosteric binding of xanomeline at the other mAChR subtypes is very interesting and should be expanded to include some mutagenesis study, thereby revealing the structural basis of subtype selectivity. Meanwhile, the protein expression for these mutants should be measured.

Minor issues:

1. The modeling of the complex in the main fits the map, while several side-chains which are modeled are not well-supported by density and should be removed, such as F34 (R chain), K147 (R chain), K177 (R chain) and F425 (R chain).

2. Add measurement unit for "root mean square deviations (RMSD) of 0.64 and 0.65 for the whole complex".

3. Abbreviations such as Ipx, DREADD, ECL, ICL and TM should be defined in the text at their first occurrence.

4. Figure 3: First letter of first word (such as xanomeline, iperoxo, unbound, bound) should be capitalized.

5. Figure 4: A new panel showing sequence alignments of the xanomeline allosteric sites residues across mAChR subtype should be included.

6. Extended Data Figure 4: Update PDB code (XXX now) for M4R-Xanomeline structure.

7. Extended Data Figure 5: Add blank between D7.32 and Calpha.

8. Figure S2 should be corrected as Extended Data Figure 2, similar changes for Table 1, Table 2.

9. Figure 4: The PDB codes for M3, and M5 mAChR subtypes should be defined in the legend. Or provide computational protocol of active state homology model.

10. The soft version should be provided.

REVIEWER COMMENTS

Reviewer #1 (Remarks to the Author)

In this manuscript, Burger et al. present evidence that the muscarinic acetylcholine receptor (mAChR) agonist xanomeline simultaneously binds to two sites on the M4 mAChR—the orthosteric binding site (as previously known) and a conserved extracellular allosteric binding site (where several other mAChR allosteric modulators are known to bind)—based on a cryo electron microscopy structure of xanomeline-M4 mAChR-heterotrimeric Gi. This finding was supported by (1) molecular dynamics simulations indicating that xanomeline spontaneously binds to the allosteric site and (2) radioligand binding experiments showing that xanomeline slows dissociation of the orthosteric antagonist [³H]-NMS from M4 and other mAChRs (thus demonstrating that xanomeline must be able to occupy the allosteric site). The authors suggest that this dual binding mode could be related to some of the distinctive pharmacological properties of xanomeline (e.g., its “wash-resistant” pharmacology and negative cooperativity with LY2033298, a positive allosteric modulator of other mAChR agonists).

The proposed mechanism is quite interesting and could be important for future drug design efforts for mAChR agonists. While interactions of mAChR orthosteric agonists with the allosteric site during the binding process have been previously proposed, this manuscript provides evidence that these interactions can be quite stable for some ligands.

We thank the reviewer for their complimentary remarks on our manuscript and taking the time to provide positive and constructive feedback. We describe below in detail how the revised manuscript aims to address their main concern(s).

My main concern is the limited amount of experimental support for the allosteric binding mode and the absence of data to show that:

(1) xanomeline can in fact simultaneously bind to BOTH the orthosteric and allosteric sites. This could be assessed with radiolabeled xanomeline (if available) based on its binding stoichiometry. If not, since there are radiolabeled allosteric ligands commercially available (e.g., [³H]-LY2119620), it should at least be possible to demonstrate that xanomeline directly competes with the allosteric ligand.

We appreciate the reviewer’s concern about additional experimental support for allosteric binding of xanomeline (addressed in further detail below in point 2), and agree that the use of radiolabeled xanomeline and/or PAM (LY21195620) may provide additional data for a direct (i.e., competitive) interaction between two ligands at the allosteric site. Unfortunately, this is not possible at this point in time for the following reasons:

- a) We do not have access to any source of radiolabelled xanomeline (it is not commercially available).
- b) Although tritiated LY2119620 is available, it is a very low affinity (μ M) binder at the ECV mAChR allosteric site (see Schober et al., 2014, Mol. Pharm. 86: 106) and thus characterised by very high non-specific binding. The only means by which

any specific binding has been observed in the past (albeit still at low levels) is via the inclusion of saturating concentrations of a high efficacy agonist to occupy the orthosteric site – and given that xanomeline would also bind to this site in addition to the allosteric site, interpretation of any data from such an experiment would remain ambiguous at best.

Nonetheless, we agree with the reviewer that our work can be strengthened by additional pharmacological experiments to support an allosteric mode of engagement of xanomeline, and have thus now included a novel variant of the classic (isotopic dilution-based) dissociation kinetic assay that addresses *both* the allosteric and orthosteric binding potential of xanomeline. Specifically, instead of using a saturating concentration (10 μM) of the competitive antagonist, atropine, to promote isotopic dilution at the orthosteric site and prevent rebinding of pre-equilibrated [^3H]-NMS as a function of time, we used xanomeline itself across a range of concentrations; to our knowledge, this has never been demonstrated before. As one may predict, low concentrations of xanomeline (e.g., 0.1 and 1 μM) were insufficient to completely prevent [^3H]-NMS rebinding but, importantly, 10 μM xanomeline was able to do so – yielding an estimate of the [^3H]-NMS dissociation rate constant that was not significantly different to that determined in the presence of saturating atropine. This finding is consistent with full occupancy by xanomeline of the orthosteric site. Increasing the concentration of xanomeline, to 100 μM , then resulted in a further reduction of [^3H]-NMS dissociation – to a similar degree as that observed for the combination of 10 μM atropine + 100 μM xanomeline. Collectively, these new results strongly indicate that xanomeline is indeed able to simultaneously bind to both the orthosteric and allosteric sites at high concentrations. We have now included these new findings (shown below) as Extended Data Figure 6 in the revised manuscript, and amended the text accordingly.

(2) binding of xanomeline to the allosteric site contributes to its pharmacology. It is obviously challenging to assess the contribution of allosterically bound xanomeline to signaling since it is also an orthosteric agonist. However, the authors mention several phenomena that appear to be testable. For example, does the F186A mutation in the allosteric site affect xanomeline's "wash-resistance"?

We agree with the reviewer that the mechanistic basis of the "wash-resistant" aspects of xanomeline's pharmacology remain to be determined. We have initiated preliminary studies of this phenomenon, but feel that it is outside the scope of the current work – especially given our findings to date. That is, using a previously established protocol (PMID:17446301) comparing the binding of [³H]-NMS at the M₄ mAChR WT to that observed at the F186A mutation, we do not see any differences in the wash-resistance, as evidenced by the change in K_d between membranes that were pretreated with xanomeline. Given this preliminary finding, and the ongoing studies of xanomeline wash-resistance as a distinct mechanism from the allosteric binding described in our current study, we have removed the wash-resistance Discussion from the manuscript altogether, but include our preliminary data below for the reviewer's benefit.

Data shown are the mean ± standard error of the measurement (one biological repeat, in duplicate).

While I do not want to burden the authors with a number of additional experiments, there is currently only one set of experiments to support the dual orthosteric/allosteric binding mode (the central claim of the paper), so I think at least one more line of evidence to solidify this claim or demonstrate its pharmacological relevance is important.

We appreciate the reviewer's comment but, with respect, wish to highlight that we actually provide at least *three* distinct, but complementary, lines of evidence for the dual orthosteric/allosteric binding mode i.e., a) the cryo-EM structure; b) the MD simulations; c) pharmacological validation, including the new 'xanomeline-alone' [³H]NMS dissociation kinetic data, in addition to the extensive mutagenesis data comparing xanomeline to the

prototypical allosteric modulator, LY2033298. We posit that this degree of characterization and validation is arguably more robust than other similar studies from the GPCR field of this type of phenomenon to date.

We have also revised the manuscript to more clearly make this distinction as follows:

“Collectively, therefore, our cryo-EM structure, MD simulations and pharmacological kinetic binding assays provide three distinct, but complementary, lines of experimental evidence that xanomeline can concomitantly occupy both orthosteric and allosteric sites at the M₄ mAChR”.

“

Minor points:

1) Could you please clarify if the orthosteric xanomeline remains stably bound to this site throughout the duration of the MD simulation runs?

Orthosteric xanomeline does remain stably bound throughout the MD simulations. We have now clarified this in the main text.

2) The authors and their collaborators have previously published MD simulations with xanomeline/M4 mAChR (e.g., ref 33). Could you please clarify whether stable binding to the allosteric site was observed in simulation there (and just not understood/commented upon), or whether simulation conditions were different in this manuscript that led to the stable allosteric site interactions observed?

The MD simulations in ref 33 used a different set of simulation conditions aimed only at investigating the interactions and effects of xanomeline bound to the orthosteric site. Simulations in ref 33 were initiated with xanomeline already bound at the orthosteric site. No additional xanomeline molecules were present in the allosteric site or solution. Therefore, no allosteric binding was observed or studied in that prior publication.

3) The authors cite previous work in the discussion showing that LY2033298 and xanomeline show negative cooperativity. However, LY2033298 interacts stably with the allosteric site in simulation with xanomeline bound to the orthosteric site (Figure 3A). The authors may want to add a statement in the discussion to clarify explicitly why these findings are not incompatible.

We thank the reviewer for pointing this out and apologise for the misunderstanding, which relates to our prior study at the M₂ mAChR, not the M₄ mAChR (Valant et al., Mol Pharm 2012). In that prior study of the M₂ mAChR, LY2033298 potentiated the binding of xanomeline (i.e., *positive binding cooperativity* – albeit modestly), but inhibited the signaling of xanomeline in both cell-based and membrane-based functional assays at high concentrations (i.e., *negative efficacy modulation*). Thus, given that the ultimate cellular effect of xanomeline is manifested at the level of response (not binding), the conclusion in our prior study at the M₂ mAChR was that the *overall* effect of LY2033298 on xanomeline was a (functionally) negatively cooperative interaction, which we

interpreted as a potential example of the phenomenon of ‘probe dependence’. In regard to the M₄ mAChR, the cooperativity between xanomeline and LY2033298 has not been characterised at the human M₄ mAChR to our knowledge. However, we *have* previously found a weak positively cooperative effect in both binding and function at the mouse M₄ mAChR between the two ligands (Suratman et al, Brit. J. Pharmacol. 2011, 162: 1659). Collectively, therefore, in terms of xanomeline *affinity*, our MD simulations are indeed consistent/compatible with the finding that LY2033298 can stabilize xanomeline binding at both the M₂ mAChR and the M₄ mAChR. The divergence noted in our prior study of the M₂ mAChR occurs at the level of functional responsiveness, which has not been investigated in the context of MD simulations.

To clarify this point, we have now re-written the relative section of the Discussion as follows:

“The allosteric binding of xanomeline may also explain the confounding interaction previously characterised between xanomeline and the PAM, LY2033298, at the M₂ mAChR⁵³. In contrast to strong positive cooperativity observed for the interaction between LY2033298 and the high efficacy agonist, oxotremorine-M, at the M₂ mAChR, the interaction between LY2033298 and xanomeline at this mAChR subtype was characterised by modest positive cooperativity at the level of binding affinity, but negative cooperativity at the level of M₂ mAChR function, i.e., xanomeline’s efficacy was abolished upon increasing concentrations of LY2033298^{53,54}. At the time, this unusual finding was interpreted as an example of ‘probe-dependence’, whereby the interaction between LY2033298 and xanomeline produced a unique, non-signalling, conformation of the M₂ mAChR. In addition, a previous study of the interaction between xanomeline and LY2033298 at the mouse M₄ mAChR revealed a weak positively cooperative effect, which was interpreted in terms of potential species variability between human and rodent mAChRs (Suratman et al., BJP, 2011). However, in light of our new structural studies and demonstration that xanomeline displays allosteric binding properties at the M₂ mAChR in addition to the M₄ mAChR, it is now possible that both of these prior findings may reflect, at least in part, an unappreciated competitive interaction between LY2033298 and xanomeline for the ECV allosteric site at both receptors”.

Reviewer #2 (Remarks to the Author):

In this manuscript, Burger et al. report the active state cryo-EM structure of the human M₄ mAChR bound to xanomeline and in complex with heterotrimeric Gi. Remarkably, the structure revealed two molecules of xanomeline bound to the receptor, one in the canonical orthosteric binding site and a second one in an extracellular vestibular allosteric site (that had been previously discovered in all muscarinic acetylcholine receptors). This novel binding mode is then investigated using molecular dynamics simulations and radioligand dissociation assays in receptor mutants.

This work provides yet another fascinating example of the wide diversity in binding modes displayed by GPCR ligands. Transient recognition by an extracellular ‘vestibule’ prior to orthosteric binding is an idea that has been going around the field for some time. While

such allosteric vestibules are now well documented, this manuscript provides the first structural evidence of simultaneous binding of an agonist to both sites. The data presented are convincing and properly analyzed. The findings are original and are presented rigorously and clearly. The computational and experimental methods are adequate. In my opinion, this work is remarkable and interesting. I particularly appreciate the thought put on the discussion about the possible affinity of the allosteric binding site.

We appreciate the reviewer's enthusiastic and supportive comments regarding the interest of our work for the field and our considerations regarding deeper implications for allosteric xanomeline binding.

Still, there are some specific points in the manuscript that I think need further clarification, particularly regarding the molecular dynamics simulations and their interpretation.

Major comments

- page 3: "The percent protonation of the dihydropyridine nitrogen atom was predicated to be approx. 47% at pH 7.4". I could not find in the manuscript how this percentage was calculated. In any case, according to these numbers, half of the ligand molecules would be neutral and presumably have a low affinity for the orthosteric site. But would this affect binding to the allosteric site? I presume this would have consequences on the pharmacological properties of xanomeline. Can the authors elaborate on this? Also, I presume the authors mean 'predicted', not 'predicated'.

The pKa of Xanomeline was predicted to be 7.3 using the ACD/Percepta software with its classic algorithm. However, the other algorithm (GALAS) in ACD/Percepta gave a prediction of 10.6. These pKa's were predicted for xanomeline in solution and do not account for changes that could be due to the local environment of the protein. As such we have moved this point from the manuscript.

- page 5: "Xanomeline bound spontaneously to the allosteric site in each of the 5 simulations ..." I am surprised that the authors could observe spontaneous ligand binding in the scale of microseconds in classical molecular dynamics simulations. Usually, enhanced sampling methods are required to reproduce such binding events, which are usually much slower. Can the authors elaborate on this point? Is there anything special about the M4 mAChR or xanomeline/LY2033298 that suggest unusually fast on-rates? Have these values been experimentally measured?

The fast binding of ligands to the ECV allosteric site of mAChRs is consistent with our previous computational work; see for example, Dror. *et. al.* Nature 2013, in which structurally diverse allosteric modulators bound spontaneously to the M₂ mAChR on a microsecond timescale. The most likely explanation for fast binding kinetics of these allosteric modulators (across multiple mAChR subtypes) is that the ECV allosteric site is very open and accessible to solvent, unlike typical orthosteric binding sites, or allosteric sites of other GPCRs that are located either within the transmembrane domain or extamembranously (Thal et al., Nature, 2018, 559: 45). This hypothesis is supported by

the fact that mAChR modulators binding within the ECV domain are invariably low affinity binders (e.g., μM range) and, indeed, prior studies have highlighted that this is due to fast binding kinetics (Lazareno and Birdsall, 1995; Mol. Pharm., 48: 362; Trankle et al., 2003, Mol. Pharm., 64: 180).

- page 5: "... the orthosteric agonist, iperoxo, interacted with the allosteric binding site only transiently... Xanomeline, unlike iperoxo, binds stably to the allosteric site of the M4 mACh..." Have the authors discussed what could be the cause of this difference? I could not find such analysis in the manuscript. Did I miss it?

With multiple iperoxo bound mAChR structures (including but not limited to PDB: 7TRK, 4MQS, 4MQT, 6OIK, 6OJK) showing iperoxo bound to the orthosteric site only, we utilised iperoxo as an 'orthosteric only' positive control. We have expanded our explanation of this observation through changing the following:

"In contrast, the orthosteric agonist, iperoxo, interacted with the allosteric binding site only transiently and for a much lower fraction of simulation time ($12 \pm 5\%$), suggesting that it cannot interact in a stable manner with the allosteric site compared to either LY2033298 nor xanomeline (**Fig. 3A**). It is possible that iperoxo does not stably interact with the allosteric site because it is smaller in size and lacks aromatics rings that are present in both LY2033298 and xanomeline. In addition, a number of iperoxo-bound mAChR structures support this finding, as these show iperoxo bound in the orthosteric site only^{19,20,31,36}."

- page 6: "... whereas an increase in the ability of xanomeline to modulate 3H-NMS dissociation was observed at Y89(2.61)A, ...". What does 'an increase in the ability... to modulate' mean? That the dissociation of xanomeline is slower in Y89(2.61)A than in other mutants? Do the authors have an hypothesis about the origin of this effect?

We apologize for the lack of clarity and have reworded this finding and provided additional explanation.

"As expected, a similar effect was observed for the well-studied PAM, LY2033298, at this mutant. A loss in xanomeline modulation was also observed at other key ECV allosteric binding site mutants Y92^{2.64}A, Q184^{ECL2}A, W435^{7.35}A (**Extended Data Fig. 7, Extended Data Table 2**). Interestingly, mutation of allosteric residue Y89^{2.61}A led to an improved ability of xanomeline to further slow [³H]-NMS dissociation, however the same effect was observed when LY2033298 was tested at this mutant. Therefore, our mutagenesis experiments further support the common ECV as the allosteric binding site for xanomeline, given that it responds in the same manner as LY2033298 to residue changes within this site".

- page 6: "... and performed an energy minimization ...". Geometry optimization of the receptor-ligand complex by energy minimization only leads to a local minimum very near the starting pose. It is not clear to me that such strategy would necessarily accommodate

xanomeline in the binding pocket of receptors of different subtypes in a native binding pose. Why didn't the authors start with computational docking experiments?

We attempted to perform docking experiments first. However, docking produced many possible binding poses with similar energy scores. It could not reproduce the allosteric xanomeline binding pose observed in the M₄ cryo-EM structure as the most favorable pose. This may be because docking does not take into account the membrane interface near xanomeline's tail. Therefore, the analysis using the minimized pose was actually more helpful to answering the question of whether xanomeline could fit similarly into the allosteric site of the other mAChR subtypes. We have amended the figure to highlight the pharmacology data first, followed by a sequence alignment of the allosteric site residues alongside the minimized binding poses. We have also restructured and amended the text to better clarify this analysis.

- page 8: "... but is most likely due to xanomeline interacting with lipids through its hexyloxy tail". In this paragraph, the authors discuss the possible binding mechanism of xanomeline and cite molecular dynamics simulation studies suggesting the existence of a channel in the transmembrane bundle to access the orthosteric binding site, or a three-step binding mode, in which xanomeline would interact with the lipid bilayer, diffuse into the allosteric site, and then move to the orthosteric site. I presume that none of this was observed in the simulations performed in this work, otherwise the authors would have discussed it. If this is the case, how was xanomeline binding to the receptor in their simulations? Directly diffusing from the solvent? Is there anything remarkable in the desolvation mechanism of the ligand? Were there any interactions of the ligand with the membrane at all? I think these are points that should be discussed in further detail.

We have performed preliminary experiments to test the wash resistance of xanomeline and its allosteric nature as a possible mechanism (see response to reviewer #1, comment 2) at the F186A mutant. Wash resistance was still observed, indicating that mutating F186 has no impact on xanomeline's wash resistance. Accordingly, we have removed the wash resistance discussion from the manuscript, although it is planned to investigate the mechanisms underlying wash-resistance in further detail in future work.

Minor comments

- page 4: "Unexpectedly, strong cryo-EM density was also observed in the common mAChR extracellular allosteric binding site". Was this density observed in the maps of other major 'good' classes? Could the authors tell?

There were no other 'good' 3D class averages.

- page 7: "Collectively, these studies illustrate the potential for developing subtype selective mAChR ligands by targeting distinct orthosteric sub-pockets that exist in a conformationally dependent manner." I think I understand what the authors mean, but the expression 'exist in a conformationally dependent manner' is not very clear. The authors are referring to different conformations observed in 'static' structures, but the reader might think of 'dynamic conformations' (i.e., different conformations as a result of inherent

protein dynamics). At least, I did in the first read. Can the authors please rephrase this argumentation?

We thank the reviewer for pointing out this lack of clarity and have rephrased accordingly.

“Collectively, these studies illustrate the potential for developing subtype selective mAChR ligands by targeting distinct orthosteric sub-pockets that exist in a conformation dependent on the orthosteric ligand. The identification of additional conformationally sensitive orthosteric sub-pockets may thus be a more general phenomenon than previously appreciated that can aid selective orthosteric drug design, but requires further structural studies with novel ligands for further validation.”

- The manuscript as written is very interesting. However, I was expecting to read something about the interactions between the receptor and the G protein. As the authors did not say anything about it, I presume there is nothing remarkable in this interface. Did the authors look at it in any detail?

We did inspect this interface as part of our structure analysis. However, the M4R-G_i interface observed in our xanomeline structure was highly similar to the receptor-G protein interface observed in our recent M4R structures (Vuckovic et al., eLife 2023). Furthermore, this interface has been analysed extensively in the recent M₁R-G₁₁ and M₂R-GoA structures (Maeda et al. Science, 2019). For these reasons we chose not to elaborate on this. However, we agree with the reviewer that this analysis is important and have referenced these studies and highlighted that nothing remarkable was observed in our structure by adding the following:

The G protein interface in our xano-M₄R-G_{i1} was similar to that observed in previous agonist-bound M₄R-G_{i1} structures, as well as other mAChR G_{i/o} complex structures^{19,20,22}.

Reviewer #3 (Remarks to the Author):

In the manuscript by Wessel and collaborators, the authors describe the cryo-EM structure of active M4 muscarinic acetylcholine receptor (M4 mAChR) in complex with the G protein and agonist xanomeline. Besides the orthosteric (acetylcholine-binding) site, xanomeline was found to bind one extracellular vestibular site with the support of the solid density. To further validate this hypothesis, they performed molecular dynamics simulations and observed the binding of Xanomeline from solution to the extracellular allosteric site. Meanwhile, radioligand dissociation experiments of both wildtype and several mutants for M4 mAChR as well as other mAChRs were adopted to support the allosterically bind at all mAChR subtypes. The work provides the molecular pharmacology of xanomeline and expand the knowledge of GPCR ligand binding , which may enable the structure-based design of novel selective mAChR ligands.

The manuscripts are well-written and has shown remarkable novelty. Give there are multiple released structures of M4 mAChR, the novelty of current manuscript mainly

focused on the “allosteric” binding site. However, current data is probably not suitable for a full manuscript (if possible, communication is perfect). Several major points as well as some minor points should be addressed before further consideration.

We thank the reviewer for their favourable comments on our manuscript and hope that we have addressed their concerns.

Major issues:

1. Clearly, the authors prefer to describe the allosteric binding instead of the allosteric modulation, indeed as the data shown. The functional link between allosteric binding and allosteric modulation (mainly as downstream signaling) is not clear, i.e., how to design or optimize these types of ligands toward better efficacy is very important and I guess several lessons could be obtained from this excellent special case. Are there any functional consequences of the F186A as well as Y92A, Q184A, W435A, and Y89A? How about their influence on endogenous agonist by xanomeline (antagonism for orthosteric binding site, positive stabilization for allosteric site?)

We appreciate the reviewers' concerns, and must state that these points have been the focus of our many musings on xanomeline function. Unfortunately, with currently available pharmacological/biochemical tools and models, the functional sequelae of concomitant binding of xanomeline are difficult to define and are, indeed, the subject of ongoing and planned future studies in our laboratory.

For instance, in terms of the use of F186A/Y92A/Q184A/W435A/Y89A mutants: We have utilised radioligand dissociation binding to verify xanomeline's allosteric nature, because a positive finding in this type of pharmacological assay can be unambiguously interpreted as evidence of an allosteric binding mode. In contrast, changes in xanomeline function at these mutants could be the result of a number of different factors, of which not all may necessarily be linked to allosterically-bound xanomeline. For example, although changes in functional activity may indeed be due to a loss of xanomeline allosteric binding as a consequence of these mutations, they may also be due to additional and/or alternative mechanisms associated with the transmission of cooperativity between orthosteric and allosteric sites; changes to the access/egress (kinetics) of orthosteric ligands; effects on receptor signaling/efficacy; biased agonism or trafficking etc., and thus require further investigation that is outside the scope of the current work (see also Nawaratne, JBC 2010; Vuckovic, eLife 2023 for some examples).

In regard to the reviewer's second point, i.e., xanomeline's allosteric influence on the endogenous agonist acetylcholine: In addition to the point made above about the potential to change the access/egress of orthosteric ligands upon ECV mutation (because they still need to transit through this ECV region to get to the orthosteric site), there is the additional complication that xanomeline itself can bind to the orthosteric site. Therefore, any changes in acetylcholine function will be manifestations of both xanomeline's allosteric and orthosteric (competitive) influence. As highlighted above, this is the subject of ongoing experiments in our laboratory. The key finding of the current work is the novel observation of concomitant orthosteric/allosteric engagement of a single GPCR by a late

stage clinical drug candidate with atypical pharmacology. It is anticipated that this will spur further studies in the field.

2. As shown in Fig 3A, LY2033298 can bind to the allosteric site in a similar manner as xanomeline, while significant difference on radioligand dissociation experiments (Fig. 3B) was observed mainly due to the fact that xanomeline can also bind to the orthosteric site. Repeating these measurements with varying concentration of atropine may draw a clearer picture of cooperativity between orthosteric binding and allosteric binding. Meanwhile, a chemical structural comparison among these three ligands (LY2033298, iperoxo and xanomeline) should be included in Fig. 3.

With respect, the experiments proposed by the reviewer (i.e., varying atropine concentrations in the kinetic assays), are not appropriate for providing new insights into cooperativity and, indeed, will actually lead to experimental artifacts for the following reason:

In order to accurately determine the dissociation rate constant of an orthosteric ligand, it is absolutely vital to ensure that the pre-equilibrated orthosteric radioligand (e.g., [³H]NMS) cannot re-associate with the receptor over the entire time course of the experiment. This can be generally achieved by two approaches **i)** 'infinite dilution' in buffer or **ii)** 'isotopic dilution' in the presence of a *saturating concentration* of another, unlabeled, orthosteric ligand (e.g., 10 μM atropine); please see Lazareno and Birdsall, 1995; Mol. Pharm., **48**: 362; Christopoulos, 2000; Curr. Prot. Pharmacol., 1.22.1. The former approach is cumbersome and unwieldy, whereas the latter approach is more convenient and, as such, is routinely utilized by most of the field. *However, the key to the isotopic dilution method is the requirement for a saturating concentration of unlabeled competitor, to absolutely ensure that radioligand re-association is prevented.* It is only under this experimental condition that any changes in radioligand dissociation by another ligand (e.g., LY2033298; xanomeline, etc.) can be confidently interpreted as evidence of an allosteric interaction. In contrast, reducing the concentration of the unlabeled competitor that is used to promote isotopic dilution *will lead to a mixed population of receptors* (simple competition based on the law of mass action); i.e., some receptors that do not have radioligand bound, but other receptors that do – this will hence introduce an experimental artifact (due to radioligand rebinding) that can be misinterpreted as a change in the dissociation rate. Indeed, this artifact can be readily observed in the new experiments we have now performed with xanomeline in our response to reviewer #1 (new Extended Figure 6): low concentrations of xanomeline (0.1 and 1 μM) are *insufficient* to prevent [³H]NMS re-association to the orthosteric site, leading to the *appearance* of a marked "slowing" in [³H]NMS dissociation, whereas what is really happening is that [³H]NMS is rebinding during the time course of the assay. The exact same result would occur if the concentration of atropine were lowered beyond that required to fully saturate the orthosteric site.

In light of the dissociation kinetic experiments that we have performed with xanomeline (in addition to the original experiments with atropine), we have also re-written the relevant

section of the manuscript to highlight these issues of isotopic dilution, and how we have addressed them to allow us to conclude an allosteric binding mode of xanomeline.

3. The allosteric binding of xanomeline at the other mAChR subtypes is very interesting and should be expanded to include some mutagenesis study, thereby revealing the structural basis of subtype selectivity. Meanwhile, the protein expression for these mutants should be measured.

We agree with the reviewer that this is an interesting finding, but feel that this is beyond the scope of the current manuscript for the following reasons: 1) Although the allosteric effect of xanomeline at mAChR subtypes (determined from the dissociation kinetic assays) was most pronounced at the M₂ and M₄ mAChRs, a comparison of the apparent (low) potencies across all 5 subtypes (where possible to determine) suggests that there is unlikely to be any *bona fide* and substantial subtype selectivity for xanomeline at the allosteric site; 2) Docking of xanomeline into the allosteric site of the other mAChR subtypes did not yield as conclusive results as those that were observed at the M₄ mAChR; 3) Ultimately, validation by cryo-EM would be a more powerful approach, but is again beyond the scope of this study. These points have been clarified in the main text.

Minor issues:

1. The modeling of the complex in the main fits the map, while several side-chains which are modeled are not well-supported by density and should be removed, such as F34 (R chain), K147 (R chain), K177 (R chain) and F425 (R chain).

We thank the reviewer for the careful inspection of our structures and agreement that the complex fits the main map. Arguably, we do not believe that stubbing residues is the best solution to this age-old problem, which has been an ongoing debate in the structural biology field for decades. In these specific examples, there is clear density for at least the beta-carbon of the residues. In addition, these residues are solvent facing. Our preference is to not stub residues under these circumstances. To make this more apparent we have clarified this in the figure legend of Extended Data Figure 3 and added an Asterix next to residues for which this pertains.

2. Add measurement unit for “root mean square deviations (RMSD) of 0.64 and 0.65 for the whole complex”.

Done.

3. Abbreviations such as lpx, DREADD, ECL, ICL and TM should be defined in the text at their first occurrence.

Done.

4. Figure 3: First letter of first word (such as xanomeline, iperoxo, unbound, bound) should be capitalized.

This has been addressed.

5. Figure 4: A new panel showing sequence alignments of the xanomeline allosteric sites residues across mAChR subtype should be included.

We thank reviewer for this suggestion and have added it as panel 4J.

6. Extended Data Figure 4: Update PDB code (XXX now) for M4R-Xanomeline structure.

We have corrected this.

7. Extended Data Figure 5: Add blank between D7.32 and Calpha.

This has been addressed.

8. Figure S2 should be corrected as Extended Data Figure 2, similar changes for Table 1, Table 2.

We thank the reviewer for picking up on these inconsistencies. We have corrected this accordingly.

9. Figure 4: The PDB codes for M3, and M5 mAChR subtypes should be defined in the legend. Or provide computational protocol of active state homology model.

The structures for M₃ and M₅ mAChR subtypes in Fig. 4 were built using homology modelling. Details have been added to the methods.

10. The soft version should be provided.

We apologise, but we do not understand what this comment is referring to (i.e., we are not clear what is meant by 'soft version').

Reviewers' Comments:

Reviewer #1:

Remarks to the Author:

The authors have thoughtfully addressed all points raised in the initial review through additional data and textual clarifications. The unanticipated and fascinating mechanisms of this ligand described in this manuscript will be a valuable contribution to the field.

Reviewer #2:

Remarks to the Author:

The authors have addressed my comments adequately. I consider that this work has the quality and general interest to be published in Nature Communications.

Reviewer #3:

Remarks to the Author:

The authors have worked hard to address issues raised and added new data that improve the manuscript. The manuscript is well organized and written, and I suggest it should be approved for publication. One minor point is the versions of software programs such as Schrodinger, Phenix, Coot and PyMOL should be provided.